# Synchronous quantitative analysis of chiral mesostructured inorganic crystals by 3D electron diffraction tomography

Jing Ai[1], Xueliang Zhang[2], Te Bai[2], Qing Shen[2], Peter Oleynikov[3], Yingying Duan[1], Osamu Terasaki [3], Shunai Che [1,2] & Lu Han [1] ✉

Chiral mesostructures exhibit distinctive twisting and helical hierarchical stacking ranging from atomic to micrometre scales with fascinating structural-chiral anisotropy properties. However, the detailed determination of their multilevel chirality remains challenging due to the limited information from spectroscopy, diffraction techniques, scanning electron microscopy and the two-dimensional projections in transmission electron microscopy. Herein, we report a general approach to determine chiral hierarchical mesostructures based on three-dimensional electron diffraction tomography (3D EDT), by which the structure can be solved synchronously according to the quantitative measurement of diffraction spot deformations and their arrangement in reciprocal space. This method was verified on two samples—chiral mesostructured nickel molybdate and chiral mesostructured tin dioxide—revealing hierarchical chiral structures that cannot be determined by conventional techniques. This approach provides more precise and comprehensive identification of the hierarchical mesostructures, which is expected to advance our understanding of structural–chiral anisotropy at the fundamental level.

In contrast to the conventional chiral crystalline materials formed by chiral inorganics or solids with chiral space groups, chiral mesostructured inorganic crystals can be inductively assembled by inorganic units with achiral space groups forming multilevel chiral spatial geometries[1–5]. The characteristic hierarchical chirality can be identified from the helically coiled crystal lattice (one primary unit) in the Ångström scale to the helical stacking of these basic units and the subsequent arrangement into higher level chirality to submicrometre and micrometre scales[6–9]. Three typical types of chiral mesostructured inorganic crystals are shown in Fig. 1c−e. Owning to the cooperative effect of the multilevel chirality, the chiral mesostructured inorganic materials exhibit outstanding structural-chiral anisotropy properties, such as chiroptical activity[3,10], photomagnetic-chiral anisotropy[11],

enantiospecific discrimination and catalysis[12,13], enantiomer-dependent immunological response[14], etc.

Understanding the unique multilevel helical arrangement of these materials is not only the key to understanding their chemical and physical properties, but also an important reference for the development of new functional materials. It is therefore extremely important to develop new structural characterization techniques for the determination of hierarchical chirality in these materials. The circular dichroism (CD) spectrum is normally used for determining chiral molecules; however, the results are indirect and inaccurate for resolving chiral mesostructures[15]. X-ray diffraction[16,17] (XRD) and modern transmission electron microscopy (TEM) techniques, including the dynamical refinement of electron diffraction data[18], convergent beam

[1]School of Chemical Science and Engineering, Tongji University, 1239 Siping Road, 200092 Shanghai, China. [2]School of Chemistry and Chemical Engineering, Frontiers Science Center for Transformative Molecules, State Key Laboratory of Composite Materials, Shanghai Key Laboratory for Molecular Engineering of Chiral Drugs, Shanghai Jiao Tong University, 800 Dongchuan Road, 200240 Shanghai, China. [3]School of Physical Science and Technology, Centre for High-resolution Electron Microscopy and Shanghai Key Laboratory of High-resolution Electron Microscopy, ShanghaiTech University, 201210 Shanghai, China. ✉e-mail: luhan@tongji.edu.cn

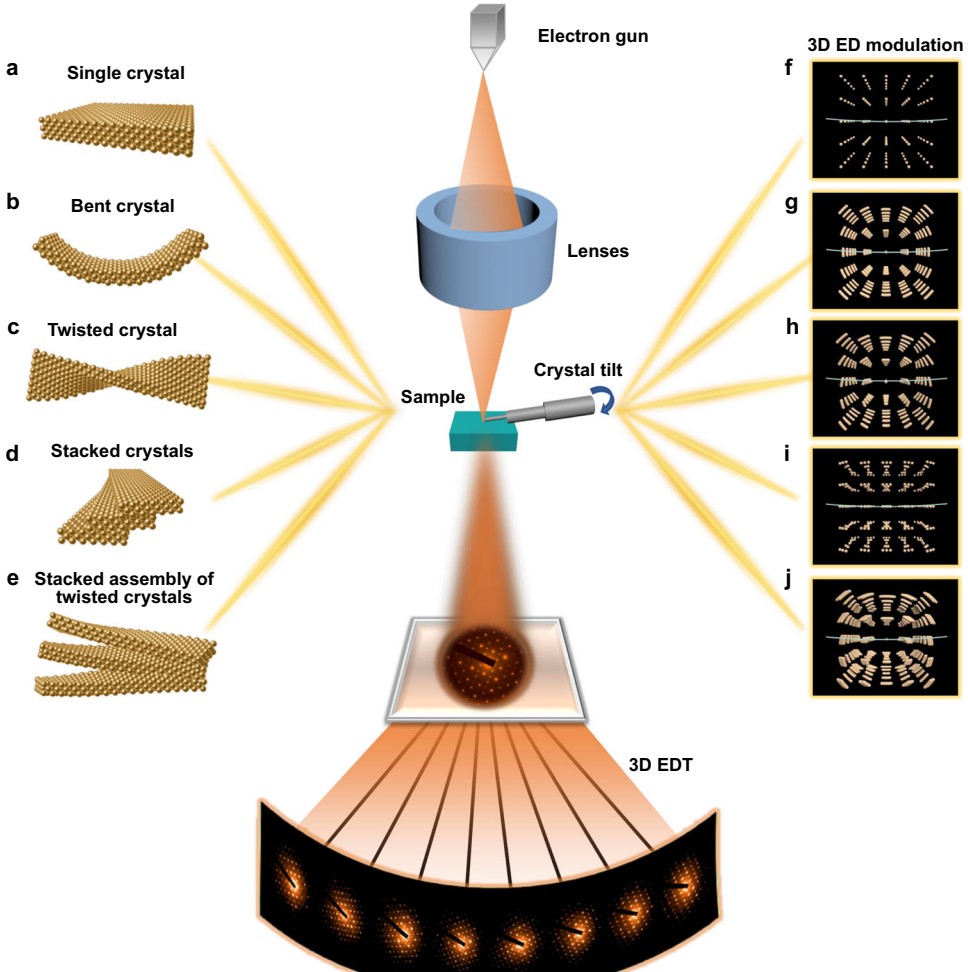

**Fig. 1 | Schematic drawing of five representative crystal structures and their corresponding modulated diffraction spots in 3D reciprocal space.** **a** single crystal, **b** bent single crystal, **c**–**e** chiral mesostructures (**c** twisted single crystal. **d** helically stacking of nanocrystals in a fan-shape. **e** stacked assembly of twisted nanocrystals). **f**–**j** the modulated electron diffraction in reciprocal space corresponding to **a**–**e**, respectively. The incident electron beam is perpendicular to the crystal surface. Both the crystal structure and the 3D ED models are shown from the same angle of view (side view). The green arc represents the Ewald sphere of the electron, which generally corresponds to the zero-order Laue zone in the conventional SAED pattern.

electron diffraction[19,20], precession electron diffraction[21], electron backscatter diffraction[22] and vortex beams[23], have been proven effective in determining the absolute configuration of crystals with chiral space groups. However, these methods are not applicable to chiral mesostructured inorganic crystals formed by achiral space groups. To resolve these mesostructures, the combination of several techniques, including CD, XRD, TEM, and scanning electron microscopy (SEM), is necessary. Although the stacking of nanocrystals and the twisted crystal morphology at the micrometer or submicrometer scales can be solved by SEM observations[15,24,25], high-resolution TEM images[26–28] and electron tomography[29,30], it is difficult to precisely determine the crystal orientation and arrangement of nanosized crystals with tiny angular variations.

The current method relies on selected area electron diffraction (SAED) by aligning the zone axis from different parts of the twisted crystal to the electron beam by goniometer tilting and the calculation of the angular deviation[31–33], the crystal orientation mapping determined by precession-assisted diffraction spot recognition[34], or the determination of the zone axis changes from different regions through nanobeam electron diffraction[35]. However, the former two methods easily lead to a deviated twisting axis, and the information is limited to two-dimensional (2D) projections of the selected area. The last approach has difficulties in determining the zone axis precisely from a

sample with a small twisting angle, and the material is vulnerable to electron beam damage.

In this investigation, we show the comprehensive structural solution of hierarchical chiral mesostructural information from the whole dataset of 3D electron diffraction (ED) reconstructed by thousands of 2D SAED patterns collected from a measurable area of the sample. As a semiautomated method to collect SAED patterns, 3D EDT[36–39] has proven to be effective for solving structures of unknown nanosized porous crystalline materials that are difficult to determine by X-ray crystallography[40–43]. This paper is based on the correspondence between the modulation of diffraction spots in reciprocal space and the mesostructure in real space and the consequent data processing.

First, we performed a Fourier transform over different modulations of the crystal structures to determine the corresponding changes in reciprocal space. Five representative types of crystal structures or assemblies based on face-centred cubic gold structures and their modulated reciprocal space are calculated and schematically illustrated in Fig. 1 (see the "Methods" section for the specific calculation operation).

i.  For a perfect single crystal without any bending or twisting, the 3D ED pattern is composed of independent diffraction spots arranged periodically in 3D reciprocal space (Supplementary Fig. 1 and

Supplementary Movie 1). In this case, the 3D ED data are analogous to the single-crystal X-ray diffraction data. The crystal structure can be solved by the unit cell parameters, the space group obtained from extinction conditions and the diffraction intensity extracted from the 3D ED data[44].

ii. Upon crystal bending, the orientation of diffracting planes changes accordingly. The reciprocal space is composed of rotational arch-shaped diffraction spots corresponding to the bending angle and orientation (Supplementary Fig. 2 and Supplementary Movie 2). The bending angle and the bending axis can be calculated by the intensity distribution of the diffraction spots in 3D. Notably, this information can be retrieved from neither the conventional SAED pattern nor the 'bending contour' in the TEM image.

iii. For individual chiral structures with twisted crystal lattices (Supplementary Fig. 3), the diffraction spots not only form an arch-shaped intensity distribution with a similar radian orientation but also represent a rotational arrangement corresponding to the torsional angle of twisted chiral crystal lattices in the characteristic direction (Supplementary Fig. 4 and Supplementary Movie 3). The deformation mode can be recognized by the rotational axis of crystal torsion and the central degree of the diffused diffraction spots with arch shapes. Notably, the difference between crystal twisting and bending can be distinguished by the arrangement of diffraction spots. The arch-shaped diffraction spots arranged in a rotational relationship suggest twisted crystal lattices, while the stretched diffraction spots without orientation change suggest crystal bending (indicated by the red arrows in Supplementary Fig. 5).

iv. For the helical stacking of nanocrystals, several sets of diffraction spots originating from each nanocrystal can be formed, following the rotational arrangement based on the stacking steps (defined as the helical stacked angle). In the overlapped part, the diffracted beam from the first crystal may be rediffracted when passing through the second crystal, revealing complicated ED patterns with satellite diffraction spots, which are responsible for the complex Moiré patterns in TEM images. Therefore, the arrangement of these rotational stacked nanocrystals can be clearly reflected in reciprocal space, where the angular or orientation relationship of these nanocrystals can be determined by the corresponding angle and orientation of these series of reflections (Supplementary Fig. 6 and Supplementary Movie 4).

v. For chiral hierarchical mesostructures consisting of primary twisted nanocrystals as the first-level chirality and the helical stacking of the primary units as the secondary chirality, the 3D ED pattern is composed of multiple datasets corresponding to the stacking of nanocrystals, and each dataset represents the arch-shaped diffraction spots arranged in rotational form due to the twisted crystal lattices (Supplementary Fig. 7 and Supplementary Movie 5). Although this information is difficult to retrieve by the conventional TEM technique, it can be observed in the reconstructed 3D ED patterns.

Consequently, the coordinate in reciprocal space corresponding to a diffraction spot $hkl$ can be transferred into Cartesian coordination $P_n$ ($X_n$, $Y_n$, $Z_n$). The vector from the origin $O$ to $P_n$ (defined as $\frac{X}{X_n} = \frac{Y}{Y_n} = \frac{Z}{Z_n}$) represents the normal direction to the lattice plane and is inversely proportional to the $d$-spacing in real space. Therefore, the arrangement of crystal lattices and dihedral angles between different lattice planes can be calculated. This relationship can be extended to explain the bent or twisted crystal lattices or the rotational stacking of nanocrystals. For instance, the helical pitch length (the distance for a helical structure makes a full turn along the torsion axis) can be precisely calculated according to the crystal torsion angle and corresponding measurable sample length (see the "Methods" section for the

calculation formula), and the internal lattice arrangement of the chiral hierarchical crystal structure can be subtly resolved. In this paper, chiral mesostructured nickel molybdate (CNM) and chiral hierarchical mesostructured tin dioxide (CTD) have been taken as examples to verify this method.

## Results

### Primary chirality with twisted crystal lattices

The enantiomeric L/D-CNMs were synthesized by a hydrothermal process using L/D-serine as the symmetry-breaking agent and structure-directing agent and Ni(NO$_3$)$_2$·6H$_2$O associated with Na$_2$MoO$_4$·2H$_2$O as the inorganic precursor. The antipodal CNMs deposited on the fluorine-doped tin oxide (FTO) substrate show a yellow-green colour with a smooth and uniform surface (Fig. 2c). The crystalline structure of D-CNM was analysed using wide-angle XRD (Fig. 2a). The reflections in the resulting spectrum are in accordance with Ni(MoO$_4$) (JCPDS card No. 86-0361) with the monoclinic $C2/m$ space group (No. 12) and lattice parameters of $a = 9.566$ Å, $b = 8.734$ Å, $c = 7.649$ Å and $\beta = 114.22°$[45]. All reflections show similar intensity to the bulk sample, and no other reflections related to impurities were detected in the XRD pattern. The SEM investigations of D-CNM present asymmetric rod-like morphology vertically grown on the substrate with a length of 800 nm–1.85 μm and a height of ~15 μm. The L-CNM exhibits a similar but mirror-imaged morphology (Fig. 2a and Supplementary Fig. 8). To determine the hierarchical mesostructure, the individual rod-like particles were first investigated using conventional TEM imaging and SAED after scraping the sample off the substrate followed by slicing of the sample embedded in epoxy resin. Low-magnification TEM images and corresponding SAED patterns taken from the side view of a D-CNM show a rod-like morphology (Supplementary Fig. 9). The SAED pattern taken from the [0$\bar{1}$0] direction reveals the single crystalline feature, and the exposed facets are (100) and (001). Notably, when the top part (indicated by the red circle) is aligned to the [0$\bar{1}$0] zone axis, the crystal orientation becomes misaligned in the middle and bottom parts. By tilting the rod-like particle along the 20$\bar{4}$ reflection, i.e., keeping the (20$\bar{4}$) plane parallel to the electron beam, by 2.3° and 6.7°, the middle and bottom parts become well aligned, respectively (indicated by the blue and yellow circles). This phenomenon demonstrates that the crystal structure is continuously twisted along the normal direction of the (20$\bar{4}$) plane in a right-handed manner with a pitch length of ~37.6 μm. Therefore, the rod-like particle can be concluded to exhibit a right-handed coiled morphology grown perpendicular to the substrate, implying the primary twisted chirality of D-CNM. However, only partial information can be obtained by this method, and it is extremely difficult to legibly distinguish chiral characteristics from 2D projections.

On the other hand, 3D EDT stands out for integrally chiral information acquisition and crystal structure determination from nano- and submicron-sized mesostructured crystals. Combining the alpha-tilting axis of goniometer from −56.9° to +61.6° (the sample located in line with the goniometer axis, equivalent to tilting the target crystal), a whole dataset of 3D ED (Fig. 2h and Supplementary Movie 6) can be built from the top area of the rod-like particle with ~2.3 μm length and 470 nm width covered by the SAED aperture (indicated by the white circle, named A$_1$ in Fig. 2g). The correspondence of the crystal orientation of the image and ED can be determined by the position of the beam stopper upon changing the diffraction focus in the diffraction mode (annotated by the same white rectangle in Fig. 2g, h, see the "Methods" section for detail). The 3D ED observed from the $\boldsymbol{a}$*-, $\boldsymbol{b}$*- and $\boldsymbol{c}$*-axes (see Fig. 2i–k and Supplementary Fig. 10) validates the unit cell parameters of $a = 9.9$ Å, $b = 8.8$ Å, $c = 7.8$ Å and $\beta = 114.7°$ (Supplementary Table 1). The reflection conditions deduced from the 3D reciprocal lattice meet the systematic extinction rules[46] of $hkl$, $hk0$, $0kl$: $h + k = 2n$; $h0l$, $h00$, $00l$: $h = 2n$ and $0k0$: $k = 2n$, suggesting the three possible space

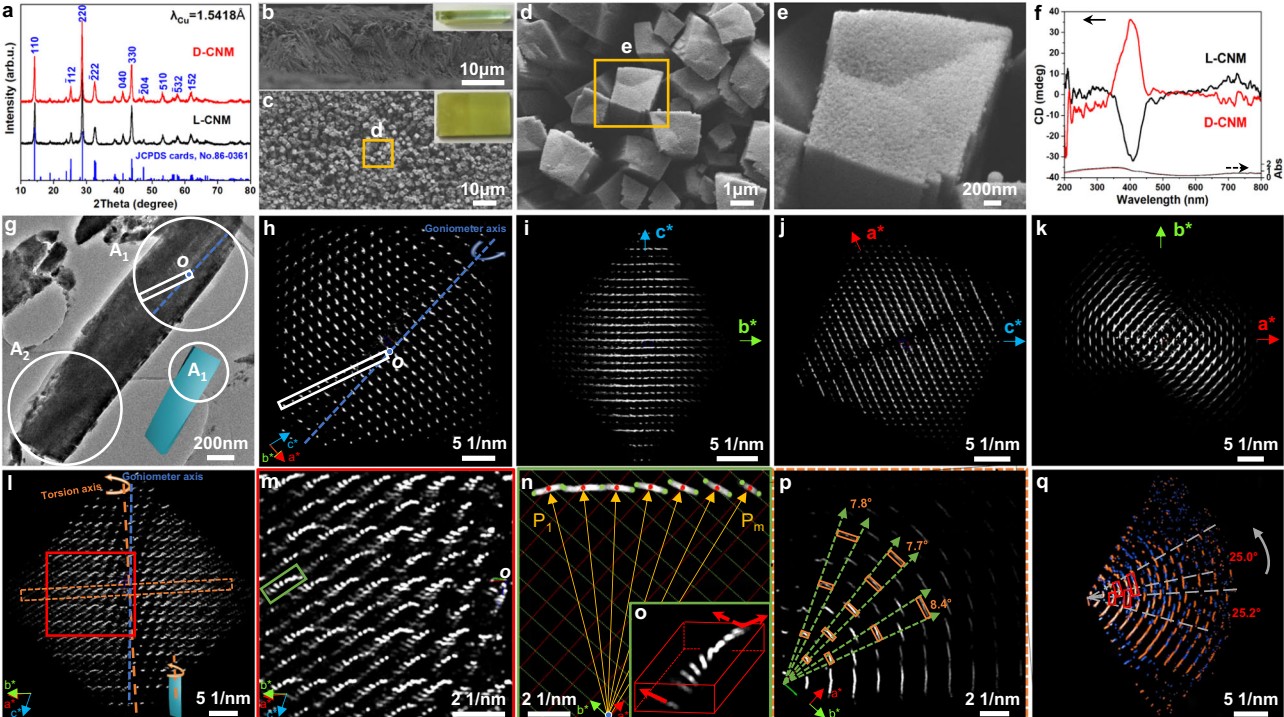

**Fig. 2 | Determination of primary chirality with twisted crystal lattices in D-CNM. a** XRD pattern of CNMs taken with Cu Kα radiation, λ = 1.5418 Å. **b, c** SEM images of D-CNM taken from the side view (**b**) and top view (**c**). Insets show the corresponding photographs. **d, e** Enlarged SEM images shown in **c** at various magnifications. **f** The DRCD spectra of the antipodal CNMs. **g, h** TEM image and 3D ED of the particle. The blue dotted line represents the position of the goniometer axis, and the white circles in **g** represent the two positions of the SAED aperture for taking the 3D ED. The top and bottom areas of the rod-like particle are denoted as $A_1$ and $A_2$. The white rectangle shows the position of the beam stopper, and *O* is the origin. The inset of **g** shows a schematic drawing of the twisted crystal. **i–k** 3D ED of the particle taken from $A_1$ observed from the ***a****-, **b****-* and ***c****-axes, respectively. The diffraction spots grayscale intensities are square roots of the corresponding measured values for a better visual experience during the publication. **l, m** The

enlargement permutation of the typical arch-shaped ED distribution cut from the reconstructed reciprocal lattice of $A_1$. The orange dotted line shown in **l** represents the position of the chiral torsion axis. **n** The localized amplification of reflection (green box in (**m**) seen from the [001] axis. $P_m$ represents the coordinates of diffraction spots (centres of the diffractions are marked by red dots), and the yellow arrows represent the normal direction of the corresponding lattice planes. Inset **o** shows the orientation deflection of 14, 0, 0 and 680 in the $h(14-h)0$ reflections. **p** Quarter amplification of arch-shaped diffraction spots of the $h\bar{h}l$ slice (orange dotted rectangle of **l**) with rotational intensity distribution perpendicular to the rotational axis of $A_1$ in the particle. **q** The superposed $h\bar{h}l$ slices from two datasets were taken from positions of $A_1$ (orange) and $A_2$ (blue) in the particle. The rotational intensity distribution of the two sets of 3D ED data illustrates lattice twisting from $A_1$ to $A_2$. Source data are provided as a Source Data file.

groups $C2$ (No. 5), $Cm$ (No. 8) and $C2/m$ (No. 12), however, the only possible space group $C2/m$ (No. 12) is chosen by comparing all possible space groups of molybdenum nickel oxide. This result is consistent with the XRD analysis, and the maximum error of the cell parameters determined by 3D EDT is 3.5% (-0.3 Å) and 0.5° for the β angle compared with the standard JCPDS card, and the error from averaged unit cell parameters calculated from all the 3D ED datasets is 1.7%. The maximum value of mean absolute deviation and standard deviation of the unit cell parameters calculated from all 3D ED datasets are 0.14 and 0.17 Å, respectively. Notably, a very complex modulated spatial arrangement of the diffraction spots can be observed, which can be interpreted by hierarchical chiral mesostructural modulation as follows.

i. The 3D ED profile shows a single crystalline-like feature (Fig. 2h–k), from which no ring pattern can be observed. Interestingly, a complex modulation of the ED in 3D can be realized. Each diffraction spot represents a group of parallel lattice planes, and the stretching of the diffraction is consistent with the angular change of the corresponding lattice planes within the selected region. In general, a thin crystal will cause the intensity distribution of diffraction spots to elongate, but not to an angular deflection. As shown in Fig. 2l, m, nearly all diffraction spots are arch-shaped with a rotational intensity distribution, implying the effect of lattice distortion. On the one hand, the stretched arcs of all diffraction spots are arranged along one

central axis, which is determined along the $20\bar{4}$ diffraction (indicated by the orange dotted line in Fig. 2l), suggesting that the crystal exhibits the most significant torsion along the normal of the $(20\bar{4})$ planes (Supplementary Fig. 11). On the other hand, the arch-shaped diffraction spots also gradually revolved perpendicular to the rotation axis along the $[1\bar{1}0]$ axis (indicated by the red arrows of Fig. 2o). It is worth noting that crystal bending often leads to the simple rotation of the diffraction spots into a concentric arrangement according to the bending axis. The rotational modulation of the diffraction spots along the two axes in the CNM can only be attributed to the continuous distortion of the crystal lattices into the chiral structure. Therefore, the modulation demonstrates the primary chirality of twisted crystal lattices in D-CNM.

ii. The torsion of the crystal lattices can be calculated by measuring the rotational angle of the arch-shaped diffraction spots perpendicular to the rotational axis. As shown in Fig. 2p, several series of diffractions present the concentric arch-shaped intensity distribution from the $h\bar{h}l$ slice. The average central angle of all visible diffractions (marked by the orange rectangles in Fig. 2p) can be determined to be 8.0°, and the pitch length is calculated to be 30.1 μm. This value can also be determined by the average value of the rotational angles calculated by the stretching end of visible diffractions, for example, the $h(14-h)0$ shown in Fig. 2n (see Supplementary Fig. 12 and Tables 2–4).

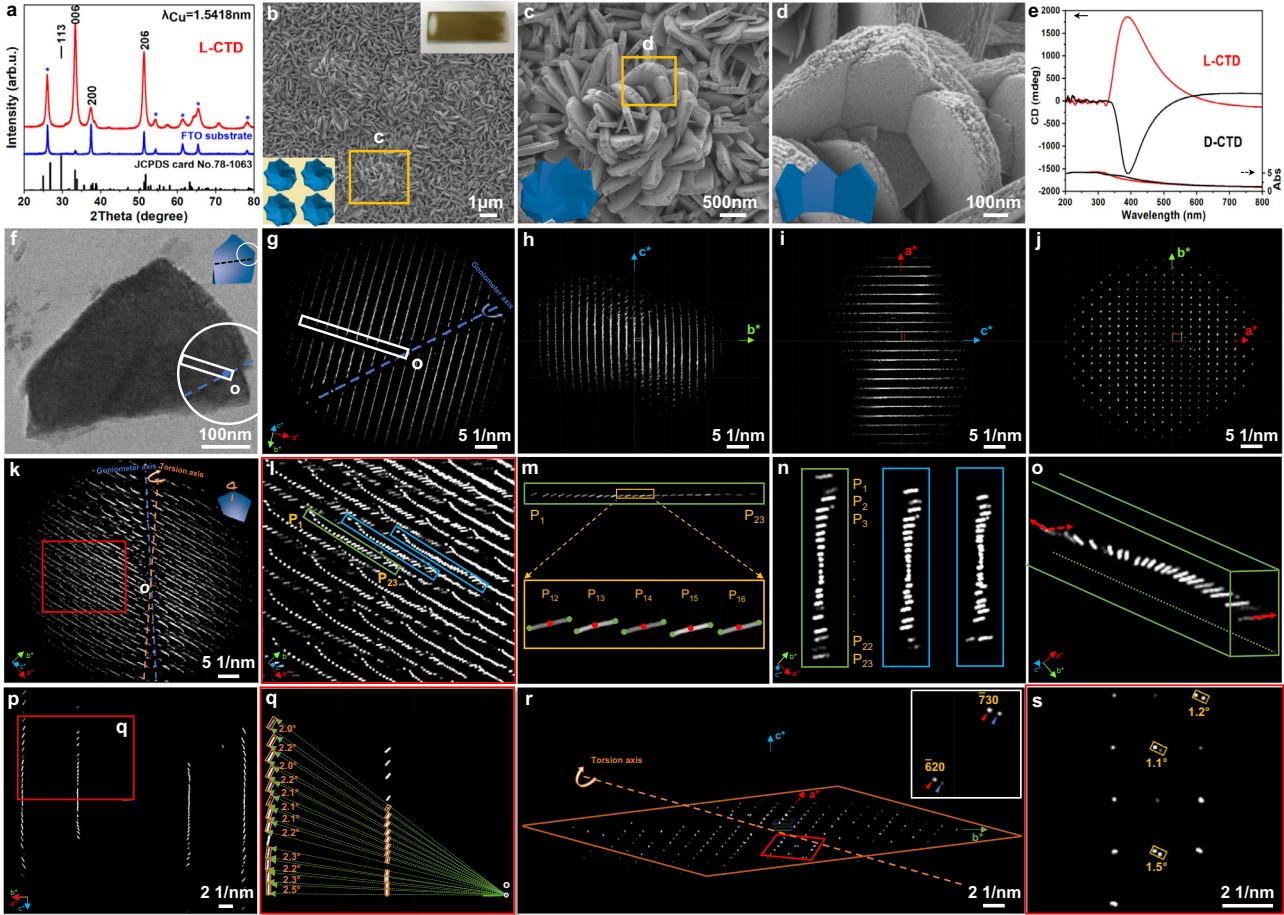

**Fig. 3 | Determination of the hierarchical chirality in L-CTD. a** XRD pattern of L-CTD taken with CuKα radiation, λ = 1.5418 Å. **b–d** Photograph (upper right inset in **b**) and SEM images of L-CTD taken at various magnifications. The inset shows the corresponding schematic structural model. **e** TCD spectra of the antipodal CTDs. **f, g** TEM image and 3D ED of the plate-like particle. The blue dotted line represents the position of the goniometer axis, the white circle (**f**) represents the SAED aperture, the white rectangle represents the position of the beam stopper, and O represents the origin. The inset shows the corresponding structural model. **h–j** 3D ED of L-CTD observed from the ***a*****-, ***b*****- and ***c*****-axes, respectively. **k** Reconstructed 3D reciprocal space, from which multiple arch-shaped diffraction spots (red box) can

be seen towards the origin. The orange dotted line represents the position of the torsion axis corresponding to the inserted model. **l, m** Step-by-step enlargement permutation of the series of arch-shaped reflections. **n, o** Typical reflections (green and blue boxes in **l**) along the vertical direction and their orientation deflection. **p, q** The amplification of arch-shaped diffraction spots with rotational intensity distribution perpendicular to the rotational axis. O represents the origin. **r** *hk*0 slice cut from the 3D ED, and the inset shows the superposition of diffraction spots. **s** The enlargement permutation of the diffraction spots (red box in **r**). Source data are provided as a Source Data file.

iii. To determine the handedness of the continuous chiral torsion, two datasets from different locations (positions $A_1$ and $A_2$ shown in Fig. 2g) of one particle were taken. The spatial orientation of the two sets of 3D ED shows a right-handed relationship from $A_1$ to $A_2$ (see the grey arrow in Fig. 2q). The pitch length calculated from $A_2$ is 39.5 μm. Consequently, the crystal torsion angle of the rod-like particle taken from D-CNM can be determined to be 25.1° along the normal direction of the (20$\bar{4}$) lattice plane (see below), and the pitch is ~33.0 μm via the corresponding rod length measured as 2.3 μm from the farthest points from $A_1$ to $A_2$ (Supplementary Fig. 13). Compared to the manual tilting of the goniometer axis, the 3D EDT method provides more intuitive evidence of lattice torsion without determining the target zone axis, and a more accurate twist angle can be calculated.

The L-CNM exhibits a similar rotational relationship with the opposite chiral direction (Supplementary Figs. 14–16). More 3D ED datasets are presented to indicate the identical twisted chirality feature of the rod-like particle in L/D-CNM (see Supplementary Fig. 17 and Table 1). It is worth noting that materials with definite spatial orientations interact preferentially with polarized electromagnetic waves,

leading to the phenomenon of optical activity (OA). OA is known to be the primary attribute of molecular and microscopic chiral materials. Therefore, the crystal lattice torsion of L/D-CNM should endow CNM with OA, which can be easily detected via diffuse reflection circular dichroism (DRCD). Figure 2f presents the DRCD spectra of L/D-CNM. The antipodal CNMs exhibit mirror-imaged OA signals based on electron transition absorption at ~410 nm, implying the photochiral selectivity of the chiral mesostructure. The primary chirality with twisted crystal lattices in CNM showing a significant chirality-dependent OA response.

**Hierarchical chirality by chiral stacking of primary units**

The second example, L-CTD, was prepared with L-ammonium tartrate as a chiral symmetry-breaking agent by a hydrothermal method[47]. The L-CTD coated on the FTO substrate shows a semitransparent yellow-brown colour with a smooth surface (Fig. 3b). It is difficult to determine the fine crystal structure solely from the XRD pattern because of the strong reflections of the substrate and the peak overlap from possible polymorph structures (Fig. 3a). The sample was determined to be mainly composed of the *Pm*-cassiterite phase [WWW-MINCRYST, OXIDE_Sn-3381, JCPDS card No. 78-1063] with the space group *Pbcn*

(No. 60) and lattice parameters of $a = 4.737$ Å, $b = 5.708$ Å and $c = 15.865$ Å[48], and the fine crystal structure could be determined via the 3D EDT method (see below).

The sample shows a uniform plate-like morphology with a parallel long edge and an acute top angle grew perpendicular to the substrate with a width of ~450 nm and a thickness of ~150 nm. The plates show a rough side surface with layered sheet-like morphologies (Supplementary Fig. 18). The individual plate-like particles were revealed after scraping the sample off the substrate followed by slicing the sample embedded in epoxy resin. The crystalline structure and chirality of the plate were initially investigated using the conventional TEM technique. However, due to the relatively small crystal size and the large pitch length, the precise determination of the zone axis by manual tilting is extremely difficult. By the SAED patterns taken from three contiguous positions of one typical plate with a side length of ~270 nm, a rotational relationship with a tiny deflection of 1.1° along the $[01\bar{1}]$ zone axis in a clockwise manner was confirmed. Furthermore, the diffused intensity and split diffraction spots can be observed for the high index reflections (the red and yellow arrows in Supplementary Fig. 19), indicating a possible distortion of crystal lattices into the crystal bending or twisting arrangement.

Accurate quantitative analysis of the chiral hierarchical structure of L-CTD was also achieved by 3D EDT. Figure 3f, g shows the TEM image of one typical plate-like particle and the corresponding 3D ED covering the collection angle from −48.3° to +51.8° (see the whole 3D ED modulation in Supplementary Movie 7). The alpha-tilting axis was set back to the initial collection angle to ensure the sample has not changed, and no obvious change in crystal shape caused by electron beam irradiation damage was observed during the data acquisition (Supplementary Fig. 20). The 3D ED observed from the $a^*$-, $b^*$- and $c^*$-axes (Fig. 3h–j and Supplementary Fig. 21b–d) suggests unit cell parameters of $a = 4.8$ Å, $b = 5.9$ Å and $c = 16.1$ Å (Supplementary Table 5), and the maximum error of cell parameters is 3.4% (~0.2 Å). The error from averaged unit cell parameters calculated from all the datasets is 1.2%. The maximum value of mean absolute deviation and standard deviation from all 3D ED data are 0.08 and 0.10 Å, respectively. The long $c$-axis of the unit cell in real space leads to very tight diffraction spots along the $c^*$-axis in reciprocal space. The reflection condition deduced from the 3D reciprocal lattice meets the systematic extinction rules[46] of $0kl$: $k = 2n$; $h0l$: $l = 2n$; $hk0$: $h + k = 2n$; $h00$: $h = 2n$ and $0k0$: $k = 2n$. Only the space group $Pbcn$ (No. 60) is reasonable out of all possible space groups of tin dioxide. Notably, the arch-shaped modulation of L-CTD in reciprocal space also implies the inherent torsional structural feature of chiral self-assembly.

i. Similar to CNM, the diffraction spots of CTD show a rotational intensity distribution towards the origin $O$ (Fig. 3k), which cannot be found in commercial achiral tin dioxide (see Supplementary Fig. 22). The central rotational axis is determined as the normal of the $(1\bar{1}0)$ plane (indicated by the orange dotted line in Fig. 3k), as the rotational intensity distribution perpendicular to the $1\bar{1}0$ reflection exhibited the most significant torsion (Supplementary Fig. 23). As shown in the enlargement permutation of Fig. 3k–m, one typical set of diffraction spots $\bar{6}\bar{6}l$ (indicated by the green box in Fig. 3l) are arranged in an anticlockwise rotational manner along the perpendicular direction ([001] axis, marked by green and blue rectangles in Fig. 3n, o), also implying the primary chirality of the crystal lattice torsion. The crystal torsional angle of the plate can also be calculated by the average central angle of diffraction spots or the stretching endpoints of each diffraction spot (Fig. 3p, q and Supplementary Fig. 24 and Tables 6–8, defined by the same methods as D-CNM). More 3D ED datasets of L-CTD are presented to exhibit the consistent structural features of the sample (Supplementary Fig. 25 and Table 5). Interestingly, the high-order diffraction spots are shifted from the original reciprocal lattice sites in 3D ED data (Supplementary Fig. 26),

which can be interpreted by the deviation from the Bragg reflection position due to the twisting of chiral crystal with continuous distortion of lattice planes (Supplementary Fig. 27). This phenomenon may also relate to the change of diffraction intensity distribution induced by local thickness variations and the effect of multiple scattering[49–51]. In addition, complex contrast can be found in the high-resolution TEM (HRTEM) image (Supplementary Fig. 28), which may be related to the dislocations caused by the distortion of the reflecting planes or the overlapping of crystals.

ii. As shown in the SEM images (Fig. 3c, d), the layered sheet-like structure can be observed from the side surface of the particle, suggesting the possibility of second-level chirality. This evidence can be found in the series of collected ED patterns (Supplementary Fig. 29), showing the superposition of ED patterns corresponding to the rotationally stacked nanosheets. In the reconstructed 3D reciprocal space, the same observation can be also verified as marked by the red and blue arrows in the inset of Fig. 3r, indicating the helical stacking of nanosheets along the [001] axis in the plate. The average central angle between the visible diffraction spots to the origin represents the rotational angle between the stacked nanosheets (Fig. 3s). In general, the diffraction spots will be elongated perpendicular to the nanosheet stacking direction; however, this phenomenon is not obvious in this sample, which may be due to the small deflection angle, and the nanosheets are closely arranged in the stacking direction analogous to a single crystal. Further evidence of the superposition of diffraction spots corresponding to the two nanosheets can be found in the TEM image and the corresponding SAED pattern (Supplementary Fig. 30). Judging from the above results, it can be speculated that the formation of nanosheets may be due to the inability of crystal lattices to withstand the stress by primary distortion of the chiral structure.

Consequently, two levels of hierarchical chirality, including the primary twisted crystal lattices of nanosheets and the secondary helical stacking of these nanosheets, exist in the plate. The average value of the nanosheet crystal torsion angle was determined to be 2.2° along the normal direction of the $(1\bar{1}0)$ plane (see below), and the pitch length of L-CTD was speculated to be 26.2 µm via the corresponding sample length measured to be ~160 nm. For the secondary chirality, the average rotational angle between the helically stacked nanosheets was determined to be 1.3°.

The transmitted circular dichroism (TCD) spectrum of L-CTD shows a positive circular dichroic signal at 320–580 nm harvesting in the hierarchical mesostructures and exhibits the main signal centred at 390 nm (Fig. 3e). These behaviours occur because the chiral centres exist in dissymmetric environments in the transitions of electronic states[52]. The TCD spectrum of CTD on the FTO substrate exhibits signals including both absorption-based and scattering-based portions. The absorption-based OAs originate from the primary crystalline chirality arising from the distorted single crystal of nanosheets in L-CTD, and the helically stacked nanoplates originate from the left-handed stacking of nanosheets beget the chiral morphology, and the chiral morphology begets the chiral interface between two media with different permittivity ($SnO_2$ and air here), resulting in selective reflection of one-handedness of circularly polarized light and transmission of the other handedness, which could contribute to scattering-based OA. It is the hierarchical chirality in the mesostructure of CTD that contributes decisively to OA signalling. These results clearly reveal the structure-activity relationship of the chiral mesostructured materials.

## Discussion

In our experiment, the chirality of the inorganic crystals mainly originated from the continuous torsion of the lattices and/or the helical

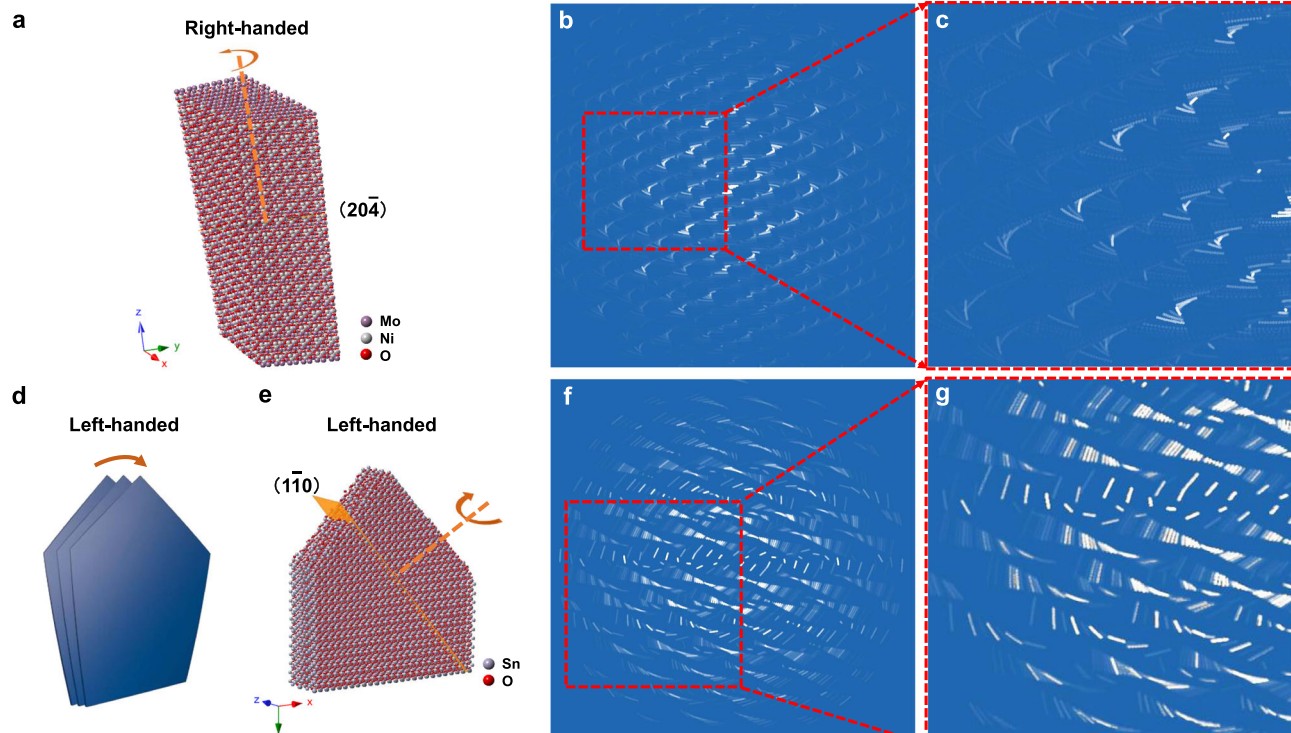

**Fig. 4 | Chiral hierarchical mesostructural model and the simulation of 3D ED.**
**a** The structural model of D-CNM. The orange dotted line represents the rotational axis, and the arrow represents the rotational direction. **b, c** Simulated 3D ED of the D-CNM crystal. Parameter setting: the number of lattices is 12, and the rotation step and twisting step are 0.6° for each lattice. **d** The schematic structural model is composed of helically stacked twisted nanosheets in L-CTD. **e** The structural model of L-CTD. The orange dotted line represents the rotational axis. **f, g** The simulated 3D ED of L-CTD. The parameter settings are as follows: the number of lattices is twenty-three, the rotation step is 0.2° for each lattice, and the twisting step is 0.15° for each lattice.

stacking of the nanocrystals. The typical morphologies of these materials are rod-type and plate-type, twisted along their axial growth direction. In the 3D EDT experiments, the particle arranged along the alpha-tilt axis of the goniometer was often chosen for the data collection to ensure the same thin area with the largest tilting angle. Sometimes this choice may cause the torsion axis close to the goniometer axis. In this case, the accurate calibration of the goniometer axis and the data alignment are the key to correctly revealing the distortion of reciprocal space. It is worth noting that the goniometer axis is mainly determined by the instrument and similar values should be obtained for both chiral and achiral samples. Besides, the diffraction spot elongation appears systematically as circular shapes around the common centre for twisting crystals, while a wrong frame alignment shifts all spots in the direction of the beam drift. These two cases can be distinguished and corrected.

Regarding the estimation of the errors, it is very difficult to measure precise values and their errors in TEMs. There are 2 angular values that have different errors: (i) the alpha-tilt angle that can be measured with the stage tilt precision (mechanical) and (ii) the 2D (in-plane) angles that can be distorted by the TEM projector lens (electronical). Practical experience shows that the error for the absolute alpha-tilt angle varies and can be as large as ±1°. For more general cases, an error of ±1.5° should be a reasonable estimation for the precision of this method. Besides, the precision can be also affected by the angle measurement error, which depends on the precision of diffraction spot positions against the central beam position. The further and sharper the spots, the higher the precision. The primary beam coordinates can be refined from the available Friedel pairs. An ED has a higher precision of the central beam position with more Friedel pairs, and the maximum error for measurement can be <0.5°.

To verify the 3D EDT method, the 3D reciprocal space of the two chiral mesostructured materials was calculated by the Fourier transform of the structural models. Because the size of the structural model is greatly limited to performing the Fourier transform in the software, the twisting angle was increased to reveal the chiral torsion within the allowed crystal size. Nonetheless, there is still a high degree of similarity between the calculated arch-shaped diffraction spots and the experimental results in D-CNM (Supplementary Fig. 31). For L-CTD, both the crystal lattice torsion and the satellite diffraction spots generated by the stacking of nanosheets are shown in the calculation and the experimental data, suggesting the consistency of our results (Supplementary Fig. 32). To fully reveal the modulation in reciprocal space, the 3D ED patterns of the two samples were simulated by the geometric relationship upon rotating and overlapping the 3D ED space by two perpendicular vectors corresponding to lattice torsion and secondary level stacking with angular steps (see the "Methods" section for detailed simulation operations). The parameters were set to experimental values. The simulated 3D EDs (Fig. 4b, c, f, g) are highly consistent with the experimental results. These results confirm the feasibility and effectiveness of 3D EDT in judging multilevel chirality and other assembled hierarchical inorganic crystals.

In summary, we show a general approach for the determination of the multilevel chirality and handedness of chiral mesostructured inorganic crystals by the 3D EDT technique. Both the first-level chirality of lattice twisting and the secondary helical stacking of the primary building blocks can be well reflected in the modulation of the reciprocal space. Upon generalizing and applying this method, a comprehensive understanding of various types of mesostructured materials with unique structural−chiral anisotropy properties can be expected. This method is not only applicable to chiral mesostructured inorganic crystals but also can be extended to other kinds of deformation of crystals with bending, coiling, etc. This project will play an important role in the structural elucidation of different chiral

mesostructured inorganic materials and other mesostructured materials in the future. It may also promote the development of new characterization methods for crystal defects and arrangements, fundamentally enhance the understanding of the structure–activity relationship of new materials, and facilitate the fabrication of new functional materials with a spatial geometric variation.

## Methods
### Preparation of CNM
In the synthesis system of CNM, the cleaned FTO layer coated on the substrate can act as seeds leading to the effective attachment of a $NiMoO_4$ film on the substrate. L/D-Serine was chosen as both the symmetry-breaking agent and the structure-directing agent for asymmetric attachment and co-self assembly with $Ni^{2+}$ ions owing to the strong chelation between carboxyl groups and $Ni^{2+}$ ions. Subsequently, it was combined with the $MoO_4^{2-}$ ions to form $NiMoO_4 \cdot xH_2O$ precipitates through a hydrothermal process. Finally, the organic components can be removed by calcination. In a typical synthesis, L-Serine (or D-serine) methyl ester hydrochloride (Balinway Technology Co., Ltd., >98%, 1 mmoL, 0.1051 g) was dissolved in distilled water (30 mL) and stirred for 30 min in a 50 mL beaker. $Ni(NO_3)_2 \cdot 6H_2O$ (Sinopharm Chemical Reagent Co., Ltd., >98%, 1 mmoL, 0.291 g) and $Na_2MoO_4 \cdot 2H_2O$ (Sinopharm Chemical Reagent Co., Ltd., >99% 1 mmoL, 0.242 g) were added to the above solution with continuous stirring for 30 min, which was then transferred to a 50 mL Teflon-lined stainless-steel autoclave. The FTO substrate was then added after being rinsed with acetone and distilled water three times in the autoclave. After heating in an electrical oven at 140 °C for a period of 12 h, the autoclave was cooled naturally to room temperature. The obtained precipitate was separated by centrifugation, and the film was rinsed several times using distilled water and ethyl alcohol. Finally, the product was dried under vacuum at room temperature overnight. Calcinations were conducted in an electrical furnace in the air at 500 °C for 6 h.

### Preparation of CTD
In the synthesis process of CTD, the tin dioxide coated on the FTO substrate could be considered as a crystal seeds layer. L-Ammonium tartrate served as chiral symmetry-breaking agent due to its strong coordinating ability between carboxyl group and divalent tin ions. The synthesis was performed in water by using $SnCl_2 \cdot 2H_2O$ and $Na_3C_6H_5O_7 \cdot 2H_2O$ as an inorganic precursor and chelate stabiliser, respectively. The chelation between $Sn^{2+}$ and sodium citrate kept the $Sn^{2+}$ from being oxidized to $Sn^{4+}$ immediately by dissolved oxygen in the ambient condition. NaOH provides the necessary alkali conditions for the formation of tin oxide crystals. After calcination, pure inorganic CTD with hierarchical chirality can be obtained. In a typical synthesis, $SnCl_2 \cdot 2H_2O$ (Sinopharm Chemical Reagent Co., Ltd., >98%, 0.90 g) and $Na_3C_6H_5O_7 \cdot 2H_2O$ (Sinopharm Chemical Reagent Co., Ltd., >99%, 2.94 g) were dissolved in distilled water (10 mL) and stirred for 30 min in a 25 mL glass vial. L-Ammonium tartrate (Sinopharm Chemical Reagent Co., Ltd., >98%, 0.626 g) was added to the above solution with continuous stirring to form a homogeneous solution (~30 min). Then, NaOH (Shanghai Adamas Reagent Co., Ltd., >96%, 10 mL) was added dropwise to the above solution with continuous stirring to form a homogeneous solution, which was then transferred to a 50 mL Teflon-lined stainless-steel autoclave. An FTO substrate was added that had been rinsed with acetone and distilled water three times in the autoclave. After heating in an electrical oven at 200 °C for a period of 12 h, the autoclave was cooled naturally to room temperature. The obtained precipitate was separated by centrifugation, and the film was rinsed several times using distilled water and ethyl alcohol. Finally, the product was dried under vacuum at room temperature overnight. Calcinations were conducted in an electrical furnace in the air at 600 °C for 6 h.

## Characterization
XRD patterns of CNM and CTD were recorded on a Rigaku X-ray diffractometer D/MAX-2200/PC equipped with Cu Kα radiation (40 kV, 30 mA, $\lambda = 1.5418$ Å) at a rate of 0.02° min⁻¹ over the range 20–80° and a Rigaku MiniFlex 600 powder diffractometer equipped with CuKα radiation (40 kV, 15 mA, $\lambda = 1.5418$ Å) at the rate of 0.2° min⁻¹ over the range 10–80°, respectively.

SEM images of CTD and CNM were obtained by a JEOL JSM-7800F. The CTD was taken with the decelerating model (GBSH) with an accelerating voltage of 6 kV and a decelerating voltage of 5 kV (landing voltage of 1 kV), and the CNM was taken with an accelerating voltage of 3 kV.

CD spectra of CTD and CNM were obtained on a JASCO J-1500 spectropolarimeter fitted with a TCD apparatus and DRCD apparatus, respectively. The scanning rate was 500 nm per minute, ranging from 200 to 800 nm.

TEM images of CTD and CNM were obtained by a JEOL JEM-F200 transmission electron microscope (point resolution 0.19 nm, lattice resolution 0.10 nm, $C_s = 0.5$ mm, $C_c = 1.1$ mm) equipped with a Schottky thermal field emission gun working at 200 kV using a double-tilt sample holder. Images were recorded on a Gatan OneView IS camera ($4k \times 4k$ pixels). The CNM samples were investigated by continuous automated data acquisition based on the goniometer-tilting scheme in a JEOL JEM-2100 Plus transmission electron microscope (point resolution 0.23 nm, lattice resolution 0.14 nm, $C_s = 1.0$ mm, $C_c = 1.4$ mm) operated at 200 kV, which reduced the acquisition time from ~1 h to 7 min. The 2ⁿᵈ SA aperture with 700 nm in diameter was chosen. Images were recorded on a TVIPS TemCam-XF416 CCD camera ($4k \times 4k$ pixels). The camera length is 600 mm and the individual frame exposure time is 500 ms. The CTD samples were studied and verified by step-wise data collection based on a combined stage-tilt/beam-tilt collection scheme in a JEOL JEM-2100 transmission electron microscope (point resolution 0.25 nm, lattice resolution 0.14 nm, $C_s = 1.4$ mm, $C_c = 1.8$ mm) equipped with a $LaB_6$ electron gun operated at 200 kV using the 3ʳᵈ SA aperture in the SAED mode with ~200 nm in diameter. The 3D ED data was collected using a combination of goniometer tilt with a step of 3° (with 0.2° overlap) combined with electron beam tilt with a rotation step of 0.1°. Images were recorded using a TENGRA CCD camera (2304 × 2304 pixels). The 3D ED data was collected by EDT-COLLECT software (www.analitex.com). The processing of raw TEM data and the determination of diffraction positions were performed by EDT-PROCESS software (www.analitex.com) and Gatan DigitalMicrograph software (www.gatan.com).

## Calculation of the helical pitch length
The helical pitch length refers to the distance for a helical structure to make a full turn along the torsion axis. It can be calculated according to the crystal torsional angle $\alpha$ in a measurable sample length $l$ as $\frac{360}{\alpha} * l$.

## Determination of relative position relationship by beam stopper
The correspondence of the reciprocal space and the real space can be revealed by changing the diffraction focus in the diffraction mode. It avoids the significant change of intermediate lens and there is no rotation between the diffraction and the image, thus the relative position of the crystal orientation and the electron diffraction can be determined by the beam stopper as a reference.

## Alignment of ED frames
In the data collection process, the central beam position is sheltered by the beam stopper to avoid electron beam irradiation damage. To align the data, the EDT-PROCESS software locates the ED pattern centre as a barycentre of Friedel pairs either automatically or by manual mode. Once the patterns are centred, the location of the projection of the tilting axis on the camera plane was determined. Each subset of 3° of the ED frames is aligned using cross-correlation and summed into a

single frame, which allows the processing software to compensate for the electron beam drift. Each summed frame was checked and visually controlled to ensure the correctness of the beam drift estimation. For the alignment of 3° sub-sets in 3D, the calculated sub-set centres (Friedel pairs) were used. After the goniometer axis refinement, the 3D ED can be visualized.

## Calculation of the reciprocal space modulation of five representative types of crystal structures or assemblies

The structural models were built by Au crystals (space group $Fm\bar{3}m$, lattice parameters $a = 4.0786$ Å). Five typical crystal structures were built, namely, a single crystal, bent single crystal, twisted single crystal, helical stacking of nanocrystals, and chiral hierarchical mesostructure with helical stacking of twisted primary nanocrystals. The bending and twisting of Au lattices were built by coordination transformation by the stepwise displacement of the Au atoms of the single crystal along the bending or twisting axis. The rotational stacked assemblies can be obtained by several Au single crystals arranged helically. The chiral stacked assembly of twisted primary nanocrystals can be built by stacking several twisted Au crystals accordingly. The 3D reciprocal space was obtained by discrete Fourier transform using the fast Fourier transform (FFT) algorithm in MATLAB software (www.mathworks.com). The atoms are treated as spheres in the calculation, and the Fourier transforms of the five representative structures are confined to the 1000 × 1000 × 1000 pixels region in 3D. The final results were visualized by Tomviz software (https://tomviz.org).

## Simulation of 3D ED of CNM and CTD

The 3D electron diffraction (ED) patterns of the two samples were simulated by software written by P.O. The 3D reciprocal space can be obtained by rotating and overlapping a series ED of a single crystal in a stepwise fashion around the fixed axes. The modulated 3D ED patterns of the twisted crystal can be simulated by rotating the ED of a single crystal by two perpendicular axes corresponding to lattice torsion and secondary level stacking with small angular steps. For example, twelve 3D ED patterns of CNM are overlapped by rotation along the normal direction of the $(20\bar{4})$ plane as the twisting axis with a step of 0.6°. Twenty-three twisted tin dioxide nanocrystals are helically stacked by 0.2° around the [001] axis and twisted by 0.15° for each crystal along the direction perpendicular to the $(1\bar{1}0)$ plane for the simulation.

## Reporting summary

Further information on research design is available in the Nature Research Reporting Summary linked to this article.

# Data availability

Source data are provided with this paper.

# Code availability

No original codes have been developed for this article. Whereas a simple Matlab programme used to compute the Fourier transforms is provided with this paper.

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

## Acknowledgements

This work was supported by the National Natural Science Foundation of China (Grant Nos. 21922304 and 21873072, L.H.; 21931008, S.C.), Fundamental Research Funds for the Central Universities (L.H.), National Key R&D Programme of China (Grant No. 2021YFA1200300, S.C.), and the science foundation of the Shanghai Municipal Science and Technology Commission (19JC1410300, S.C.) and Centre for High-resolution Electron Microscopy, SPST, ShanghaiTech University (EM-02161943, P.O. and O.T.).

## Author contributions

L.H. conceived the idea and led the project. J.A. performed the measurements of the samples by SEM, TEM, XRD and CD. T.B. and Q.S. synthesized the CNM and CTD, respectively. Y.D. and S.C. contributed to the synthesis mechanism and optical properties of the materials. J.A., X.Z., O.T., L.H. contributed to the structural analysis. P.O. wrote the 3D EDT and the simulation programmes. J.A., O.T., S.C. and L.H. contributed to the preparation of the manuscript.

## Competing interests

The authors declare no competing interests.
