## [Peer Review File · Nature Communications]

Synchronous Analysis of Chiral Mesostructured Inorganic Crystals Using Three-dimensional Electron CrystallographyREVIEWER COMMENTS

Reviewer #1 (Remarks to the Author):

The paper by Ai et al. is an interesting and well written paper on the determination of the chiral mesostructured using a combination of 3D ED and pattern simulation. The results of the paper are strongly supported and the method is very effective. This referee suggests the publication of the paper with minor revisions

Remarks:

- 1) The authors call the electron diffraction method they used 3D EDT. Since the 2D EDT does not exist I think this acronym is incorrect. It would be better just to speak about 3D electron diffraction (3D ED)
- 2) In the introduction it should be mentioned that the absolute crystal structure of a chiral compound can be determined by dynamical refinement of 3D ED data. Examples of such results can be found in Brazda et al. *Science* (2019) 364, 667.
- 3) In the introduction when the authors mention the possibility to use SAED for determining the chirality of twisted crystals they should quote the method of orientation and phase mapping obtained by scanning a parallel beam over an area and simultaneously recording the diffracted patterns. See Viladot et al. *Journal of microscopy* (2013) 252, 23.
- 4) In several places in the paper the authors speak about the pitch length but they have never defined what it is. Can you please give a definition of it?
- 5) I have noticed that the torsion axis is always very close to the experimental rotation axis. This can cause errors in its precise determination and also in the quantification of the torsion angles since errors in the determination of the rotation axis position and on the pattern centering can correlate with the spot rotation detected in the patterns. Have the authors check that? And consequently can the author give an error for the estimation of the rotation and torsion angles?
- 6) At page 3 the authors state that the determined extinction rules "suggest the only possible space group of C2/m". Actually the extinction rule determine only the extinction symbol C1_1 and cannot distinguish between the three space groups compatible with them : C2 Cm C2/m
- 7) This referee does not understand how the position of the beam stop can give the relative orientation between real and reciprocal space of the crystal. Between image and diffraction in a tem there is a rotation and without precising calibrating it nothing can be said about the relative orientation of the two spaces. Can you please better explain that?
- 8) In both analyzed samples the authors report the unit cell parameters without errors and then determine the error by comparing them with x-ray data. It would be more correct to report the usual error they have in unit cell measurement in 3D ED data instead of reporting unit cell values with 3 significant digits after the comma, a too high precision.
- 9) When the authors speak about the double diffraction in case of L-CDT sample it is not clear if it is really double diffraction or just superposition of two different rotated patterns. In case of double diffraction there must be reflections that cannot be indexed by simply rotating the original pattern. Is this the case? This should be better clarified.
- 10) In page 4 right column the authors speak about " SAED patterns taken from five continuous positions" is the sentence correct? Maybe the authors meant contiguous positions?

Reviewer #2 (Remarks to the Author):

I enjoyed reading the manuscript "Synchronous Analysis of Chiral Mesostructured Inorganic Crystals Using Three-dimensional Electron Crystallography" written by Jing Ai et al. I think it is important leap forward both for the field of application of the electron diffraction tomography and characterization of mesostructured chiral crystals. The article is written in very logical and straight-forward way and provides good and valuable data. Furthermore, the graphics and simulations are wonderful! To the best of my knowledge, this is a pioneering work on this subject, therefore, I recommend publishing it after revision.

Here are my comments:

a) General remarks:

1) References 41 and 42 – does not sound very professional to me. “Double diffraction” phenomenon and satellite reflections cannot be described and explained professionally in the textbook aimed for undergraduate students, written by Williams & Carter (by the way, in your citation you listed one author, while this book was written by two authors and the second is C. Barry Carter).

While talking about effect of structural defects on electron diffraction patterns – authors refer to a chapter, which I specifically looked for and apparently it explains nicely what structural defects exist in the crystal and provides 1 example of the effect of stacking faults on powder XRD – which is irrelevant to current study. Therefore, instead of this reference author should cite something more relevant. For example, there is a “bible” of analysis of structural defects by TEM written by JW Edington “Practical Electron Microscopy in materials science” (1976). This book also has wide explanation of satellites reflections and double diffraction phenomenon – which should be reviewed by authors, since I believe that double diffraction is not the effect seen on their patterns.

2) In Table 1 in extended data and throughout all article authors provide lattice parameters with 3 digits after the dot. Is 0.001A really the accuracy of the determination of lattice parameters using EDT? If you state such an accuracy – please look at your own results, where you wrote that for L-CNM from datasets/crystals 1 and 2 you got $a=9.535\text{A}$ and $a=9.941\text{A}$ (for example). If you consider them as being the same, the accuracy is 0.4A. So, why you put everywhere 3 digits after the dot? Same goes about the two digits after the dot in the angles.

3) The general question of accuracy/tolerance/error of measurements/method is left unanswered. In my opinion, one cannot suggest a method without evaluation of its precision. What the accuracy in determination of the twist angle, pitch length etc` is? It is of high importance since the assessment of these values is the major discovery of this article. You state in most placement “approximately”. In my view some small discussion and evaluation of the precision should be made. My guess will be that the angle determination precision is not better than 1.5-2 degrees.

4) Beam stopper was used during the acquisition of all datasets. To the best of my knowledge, usage of a beam stopper, makes it more difficult to merge the frames. This might lead to distorted reflections, sometimes even reflections which look a little bit elongated and bent. Here, appearance of the reflections is of an utmost importance. Please refer to this problem since I am speaking out of experience of having hard time merging the data with the beam stopper.

5) Methodology: not much information is given about the process of acquiring the EDT data. Was it continuously? Was it step-wise? If step wise – what was the step value? Was it in STEM mode? In TEM? Using nanobeam? Using the selected area aperture? Size of the aperture?

6) While working in the TEM and taking the EDT datasets – the crystal heats up. So, there are effects of twisting that you are looking for, but there are also bending of the crystal just because it is on the carbon film and not planar, in addition, it bends during the TEM analysis due to heating. These effects were ignored by the authors. Please elaborate.

b) Specific points:

1) Page 3, left column, bottom lines– estimation of the space group of the crystal d-CNM. According to the stated extinction conditions, the only conclusion which can be made is that the extinction (diffraction) symbol is C1-1, meaning that 3 space groups can be assigned to this data: C2 (number 5) (full symbol C121); Cm (number 8) (full symbol C1m1) and C2/m (number 12) (full symbol C12/m1). This information can be found in the International Tables of Crystallography volume A, which, by the way, was not cited and should be cited both here and in the second place, where the space group Pbcn was estimated (lucky for authors the Pbcn space group is indeed the only one related to the Pbcn extinction symbol ☺). So, summarizing my remarks: a) cite international tables in 2 places; b) show that there are in fact three options which comply with obtained extinction conditions; c) explain why you have chosen C2/m out of these three options. Even if it was not in scope of your research – it must be done correctly.

2) Page 4 bottom line in the left column vs midsection in the right column: I would expect to find in these two places the same lattice parameters, but they are different. Is the right one from the EDT and the left one from the JCPDF card?

3) Page 4, right column, line 15 from the top – the precise determination of the angle deviation (such as 0.49 degrees, 0.55 etc`, stated in the extended data) is far beyond the capability of TEM, to the best of my understanding. I don't know which size of the selected area aperture was used there

(extended data fig 19 a), but assuming you draw the exact size of the aperture, you have many overlapping position-wise. Your pattern contains averaged reflections` intensities originated from the same position with additions. In some places more inclination, since you are taking less of the “overlapped place”. Going back to the accuracy – 0.49 degrees... Please explain since analysis of intensity here does not apply! Furthermore, in any crystal that I work with (which are not chiral and not twisted) I see this effect due to natural crystal/sample bending.

4) Page 4 bottom part of the right column – I strongly disagree with the authors. I recommend them to check the book of Edington, which I referred to earlier in my report. Dislocations – will not produce any effect on the ED pattern. So, they are irrelevant to the change in the appearance of reflections. Second – from this one HRTEM image – we cannot conclude what exactly we see here, especially since we are looking into crystal, which is twisted due to chirality, as you state. May be these effects, which you point to in fig 26 in extended data, regard to off-zone axis conditions? Or local bend/strain? Or maybe local thickness variations? Stacking faults are seen nicely when they are “edge on” at specific orientation at bright field and/or dark field low magnification TEM images, see book written by Edington. Furthermore, the reflections will not move due to structural defects, they might provide satellites in specific cases. Please revise this part.

5) Page 5 – reference to double diffraction phenomena. Will the double diffraction spots move? It seems more like additions, originated by the layer beneath/above the original crystal which is twisted/shifted (i.e., not at the exact zone axis conditions) and, therefore, peripheral additional reflections shift as a function of misalignment with the upper/lower layer/crystal. Most likely the patterns are dynamical, but these reflections are not necessarily originated by double diffraction. I could not find which original reflection duplicates to reproduce itself at the position in the pattern which you mark. I want to emphasize that I do not argue with your results – the satellites due to twist/shift are clearly seen – I argue with the reason of their appearance.

Reviewer #3 (Remarks to the Author):

The paper presents important news of the structural characterizations of chiral mesostructured inorganic assemblies, which is particularly challenging and of great importance. The highlight of this paper is the full analysis of reciprocal and real space data that enables the determination of the mesostructure that is almost impossible to be solved by conventional techniques. The TEM analysis is state-of-the-art and the discussion related to chiral arrangement is very solid. From my point of view this work has good novelty and is groundbreaking, suitable for publication in Nature Communications after some minor revisions.

1. Relatively little information was provided for the materials employed in this investigation. The rationale for the synthesis of the chiral mesostructures can be included in the experimental section or in the supplementary information.
2. Add scale bar for the electron diffractions shown in Fig. 2 and Fig. 3.
3. In Fig. 2 and Fig. 3, “The amplitude of the diffraction is set to be square root for visualization.” Please clarify whether the display mode of the data affects the position judgment of the diffraction intensity.
4. In the conclusion section, the authors should state the types of materials this method is able to resolve.

Reviewer #4 (Remarks to the Author):

Overall this is a nice paper. I am not an expert in the area of Chiral crystals, so I can only look at the work in a general sense. I see nothing technically wrong with the work. However, the authors fail to

make any case for it to appear in NatComm; as written it is much more appropriate for Acta Cryst or Ultramicroscopy. Main comments, including more on the scope of the paper are:

1) It took me a few moments before I understood the focus of the paper, for which Fig. 1 is very useful. I think the first paragraph which start "In contrast.." needs to refer to Fig. 1 mentioning the different types of crystals present. This will significantly help the reader, who otherwise has to wait until the second page.

2) I am horrified by the last sentence of the paper:

"This project may promote the study of electron crystallography and provide significant opportunities in developing new chiral materials."

There is nothing of sublime importance in electron crystallography, nobody will get a Nobel for it by itself. What matters is the science that it can reveal.

3) Related to point 2), as it stands this paper should appear in either Acta Cryst or Ultramicroscopy, as it is all technique. The authors have to make the case for the larger science. For instance, are their chiral materials used in biology or applications where the macro-structure is unknown? I suspect that they can, they will have to in a revised version.

Point-by-point response to the reviewers' comments

Reviewer #1 (Remarks to the Author):

The paper by Ai et al. is an interesting and well written paper on the determination of the chiral mesostructured using a combination of 3D ED and pattern simulation. The results of the paper are strongly supported and the method is very effective. This referee suggests the publication of the paper with minor revisions

Remarks:

1) The authors call the electron diffraction method they used 3D EDT. Since the 2D EDT does not exist I think this acronym is incorrect. It would be better just to speak about 3D electron diffraction (3D ED)

Answer: Thank you very much for your suggestions. The concept of 3D electron diffraction tomography (3D EDT) was proposed for collecting 3D electron diffraction data in a computer-controlled TEM (Gemmi, M. *et al.* *Z. Kristallogr.* **228**, 51-58 (2013)). The word “tomography” is important in describing the method due to the reconstruction of 3D reciprocal space from individual ED patterns. The acronym “3D EDT” was used in many publications (Liu, Y. *et al.*, *Science* **351**, 365–369 (2016); Ma, Y. *et al.*, *Nat. Mater.* **16**, 755–759 (2017); Shen, X. *et al.*, *Angew. Chem. Int. Ed.* **57**, 724–728 (2018), etc.) to refer to the method. Nowadays, the “EDT” abbreviation covers all currently available methods, which can be different in the implementation. Therefore, we think it is better to keep the acronym “3D EDT” for the method so it can be tracked in the literatures. However, we can use the term “3D ED data” for the dataset obtained by the 3D EDT method. We have revised the manuscript accordingly in the main text and the Methods section.

2) In the introduction it should be mentioned that the absolute crystal structure of a chiral compound can be determined by dynamical refinement of 3D ED data. Examples of such results can be found in Brazda et al. *Science* (2019) 364, 667.

Answer: Thank you for your suggestion. We have revised the introduction of the manuscript to include the dynamical refinement of 3D ED data. “X-ray diffraction (XRD) and modern transmission electron microscopy (TEM) techniques, including the dynamical refinement of electron diffraction data¹³, convergent beam electron diffraction, precession electron diffraction, electron backscatter diffraction and vortex beams, have been proven effective in determining the absolute configuration of crystals with chiral space groups”, and added the corresponding reference (ref 13).

3) In the introduction when the authors mention the possibility to use SAED for determining the chirality of twisted crystals they should quote the method of orientation and phase mapping obtained by scanning a parallel beam over an area and simultaneously recording the diffracted patterns. See Viladot et al. *Journal of microscopy* (2013) 252, 23.

Answer: Thank you for your suggestion. We have revised the second paragraph on the left column of the first page and added the corresponding reference (ref 29). “The current method relies on selected area electron diffraction (SAED) by aligning the zone axis from different parts of the twisted crystal to the electron beam by goniometer tilting and the calculation of the angular deviation, the crystal orientation mapping determined by precession-assisted diffraction spot recognition²⁹, or the determination of the zone axis changes from different regions through nanobeam electron diffraction”.

4) In several places in the paper the authors speak about the pitch length but they have never defined what it is. Can you please give a definition of it?

Answer: Thank you for your comments. The helical pitch length refers to the distance for a helical structure makes a full turn along the torsion axis. It can be calculated according to the crystal torsional angle α in a measurable sample length l as $\frac{360}{\alpha} * l$. We have added the definition in the main text and the calculation formula in the Methods.

5) I have noticed that the torsion axis is always very close to the experimental rotation axis. This can cause errors in its precise determination and also in the quantification of the torsion angles since errors in the determination of the rotation axis position and on the pattern centering can correlate with the spot rotation detected in the patterns. Have the authors check that? And consequently can the author give an error for the estimation of the rotation and torsion angles?

Answer: In our experiment, the chirality of the inorganic crystals is mainly originated from the continuous torsion of the lattices and/or the helical stacking of the nanocrystals. The typical morphologies of these materials are rod-type and plate-type, twisted along their axial growth direction. In the 3D EDT experiments, the particle arranged along the alpha-tilt axis of the goniometer was often chosen for the data collection from the same thin area covered by the SAED aperture with the largest tilting angle while avoiding possible overlapping of the selected area with other particles. Sometimes this choice may cause the torsion axis close to the goniometer axis and sometimes these two angles are quite different. As shown in Figure R1, the torsion axis is along the normal of the $(1\bar{1}0)$ planes in the L-CTD sample, while the goniometer axis is close to the normal direction of the $(1\bar{1}1)$ planes, leading to a dihedral angle of 16.8° .

We agree that the torsion axis close to the experimental goniometer axis may cause errors in the precise determination and quantification of the torsion angles. It is worthy to note that the calibration of the goniometer axis is the key for obtaining the correct 3D ED data. The goniometer axis is mainly determined by the instrument and similar values were obtained from both chiral and achiral samples. Therefore, the distortion of reciprocal space can be correctly revealed by the reconstructed 3D ED data.

Regarding the estimation of the errors, it is very difficult to measure precise values and their errors in TEM. There are 2 angular values that have different errors: (i) the alpha-tilt angle that can be measured with the stage tilt precision (mechanical) and (ii) the 2D (in-plane) angles that can be distorted by the TEM projector lens (electronical). The practical experience shows that the error for

the absolute alpha-tilt angle varies and can be as large as $\pm 1^\circ$ although the TEM manufacturers may claim the tilting precision of 0.1° or even 0.05° , however, these small values are not related to the absolute angle value measurements. We have added the discussion of error estimation in the Discussion part of the manuscript.

Figure R1. The reconstructed 3D ED data for L-CTD sample. The chiral torsion axis (red line) and the goniometer axis (blue line) were marked.

6) At page 3 the authors state that the determined extinction rules “suggest the only possible space group of $C2/m$ ”. Actually the extinction rule determine only the extinction symbol $C1_1$ and cannot distinguish between the three space groups compatible with them: $C2$ Cm $C2/m$

Answer: We apologize for the ambiguous descriptions. We agree that the $C2/m$ cannot be determined uniquely as there are three possible space groups $C2$ (No. 5), Cm (No. 8), $C2/m$ (No. 12) matching the extinction conditions. The possible structural forms of $Ni_xMo_yO_z$ found in Materials Project Database (<https://materialsproject.org/>) or Inorganic Crystal Structure Database (<https://icsd.fiz-karlsruhe.de/search/>) are $NiMoO_4$ ($P2/c$), $NiMoO_4$ ($C2/m$), $Ni_2Mo_3O_8$ ($P6_3mc$), $NiMo_4O_{15}$ ($P1$), $Ni_4Mo_4O_{19}$ ($P1$), $NiMoO_4$ ($C2/c$) and $NiMoO_3$ ($Pm3m$). Wide-angle X-ray diffraction confirmed that the reflections are in accordance with $Ni(MoO_4)$ (JCPDS card No. 86-0361) with the monoclinic space group of $C2/m$ (No. 12) and lattice parameters of $a = 9.566 \text{ \AA}$, $b = 8.734 \text{ \AA}$, $c = 7.649 \text{ \AA}$ and $\beta = 114.22^\circ$. Therefore, the only possible configuration of nickel molybdate crystal structure is $C2/m$ (No. 12). To clarify, we have revised the sentence into “The reflection conditions deduced from the 3D reciprocal lattice meet the systematic extinction rules of hkl , $hk0$, $0kl$: $h + k = 2n$; $h0l$, $h00$, $00l$: $h = 2n$ and $0k0$: $k = 2n$, suggesting the three possible space groups $C2$ (No. 5), Cm (No. 8) and $C2/m$ (No. 12), however, the only possible space group $C2/m$ (No. 12) is chosen by comparing all possible space groups of molybdenum nickel oxide”.

7) This referee does not understand how the position of the beam stop can give the relative orientation between real and reciprocal space of the crystal. Between image and diffraction

in a TEM there is a rotation and without precisely calibrating it nothing can be said about the relative orientation of the two spaces. Can you please better explain that?

Answer: Thank you for your comments. In our experiments, the correspondence of the reciprocal space and real space can be revealed by changing the diffraction focus in the diffraction mode. This method avoids the significant change of intermediate lens and there is no rotation between the diffraction and the image. Although this “image” in diffraction mode can be very defocused, the relative position of the crystal orientation and the electron diffraction can be well determined by the beam stopper as a reference in the photograph. To clarify, we have added these discussions in the manuscript and the Methods part.

8) In both analyzed samples the authors report the unit cell parameters without errors and then determine the error by comparing them with x-ray data. It would be more correct to report the usual error they have in unit cell measurement in 3D ED data instead of reporting unit cell values with 3 significant digits after the comma, a too high precision.

Answer: We agree. In a TEM, the errors can be caused by many parameters, such as distortion of projector lens, the error in the alpha-tilt measurement and the specimen height. The errors in the unit cell parameters determined by the TEMs are usually compared to the standard JCPDS card corresponding to XRD results. (Sun, Q. *et al.*, *J. Mater. Chem. A*, **2**, 17828–17839 (2014); Gemmi, M. *et al.* *Z. Kristallogr.* **228**, 51-58 (2013); Kolb, U. *et al.*, *Cryst. Res. Technol.* **46**, 542–554 (2011)). In the previous manuscript, we used the unit cell parameters given by the software and the precision exceeded the capability of TEM. Herein, we have modified the listing of the unit cell parameters determined by the 3D EDT method with one digit after the decimal point. Taking D-CNM (A_1 , the top area of the rod-like particle) as example, the unit cell parameters were calculated using EDT-PROCESS software, showing a maximum error of 3.5% (~ 0.3 Å) from the theoretical unit cell parameters and a maximum deviation of 0.5° from the ideal β angle of 114.22° compared with the standard JCPDS card. The maximum value of mean absolute deviation and standard deviation of the unit cell parameters calculated from all 3D ED datasets are 0.14 Å and 0.17 Å, respectively. Similarly, the maximum error in unit cell parameters of L-CTD determined by 3D EDT is 3.4% (~ 0.2 Å) and the maximum value of mean absolute deviation and the maximum standard deviation from all 3D ED data are 0.08 Å and 0.10 Å, respectively. In addition, the error from averaged unit cell parameters calculated from all the datasets is 1.7% for CNM and 1.2% for CTD, suggesting the error can be effectively reduced to within 2% by averaging multiple datasets. The corresponding discussions have been added in the main text and Supplementary Information Table 1 and Table 5, respectively.

Table R1. Lattice parameters of CNM

	a	b	c	β	Space group
D-CNM (A_1)	9.9 Å	8.8 Å	7.8 Å	114.7°	C2/m (12)
D-CNM (A_2)	9.6 Å	8.9 Å	7.7 Å	114.2°	C2/m (12)
D-CNM ₁	9.9 Å	8.7 Å	8.0 Å	114.4°	C2/m (12)
L-CNM (B_1)	9.6 Å	8.9 Å	7.7 Å	114.2°	C2/m (12)
L-CNM (B_2)	9.7 Å	8.9 Å	7.7 Å	114.3°	C2/m (12)
L-CNM ₁	9.5 Å	8.6 Å	7.6 Å	114.2°	C2/m (12)
L-CNM ₂	9.9 Å	8.8 Å	7.8 Å	114.8°	C2/m (12)

Table R2. Lattice parameters of L-CTD

	a	b	c	Space group
L-CTD	4.8 Å	5.9 Å	16.1 Å	Pbcn
L-CTD ₋₁	4.8 Å	5.8 Å	16.0 Å	Pbcn
L-CTD ₋₂	4.8 Å	5.7 Å	16.0 Å	Pbcn
L-CTD ₋₃	4.7 Å	5.7 Å	16.0 Å	Pbcn

9) When the authors speak about the double diffraction in case of L-CTD sample it is not clear if it is really double diffraction or just superposition of two different rotated patterns. In case of double diffraction there must be reflections that cannot be indexed by simply rotating the original pattern. Is this the case? This should be better clarified.

Answer: Thank you. By checking the SAED data, we agree that the observed phenomenon is the superposition of the rotation of two diffraction patterns in L-CTD sample. As shown in Figure R2, two sets of diffractions corresponding to the two nanosheets with a rotational central angle of 4.1° can be observed. Accordingly, we have revised the corresponding part of the manuscript on page 5, left column and added Figure R2 as Supplementary Information Fig. 29.

Figure R2. HRTEM image and the corresponding SAED pattern taken from [001] zone axis of the plate-like particle in L-CTD. a, The TEM image of overlapped two nanosheets. **b,** The corresponding SAED pattern. The average central angle (4.1°) between the visible diffraction spots to the origin represents the rotational angle between the two nanosheets.

10) In page 4 right column the authors speak about “SAED patterns taken from five continuous positions” is the sentence correct? Maybe the authors meant contiguous positions?

Answer: Thank you for your suggestion. The word “continuous” has been changed to “contiguous”.

Reviewer #2 (Remarks to the Author):

I enjoyed reading the manuscript “Synchronous Analysis of Chiral Mesostructured Inorganic Crystals Using Three-dimensional Electron Crystallography” written by Jing Ai et al. I think it is important leap forward both for the field of application of the electron diffraction tomography and characterization of mesostructured chiral crystals. The article is written in very logical and straight-forward way and provides good and valuable data. Furthermore, the graphics and simulations are wonderful! To the best of my knowledge, this is a pioneering work on this subject, therefore, I recommend publishing it after revision. Here are my comments:

a) General remarks:

1) References 41 and 42 – does not sound very professional to me. “Double diffraction” phenomenon and satellite reflections cannot be described and explained professionally in the textbook aimed for undergraduate students, written by Williams & Carter (by the way, in your citation you listed one author, while this book was written by two authors and the second is C. Barry Carter).

While talking about effect of structural defects on electron diffraction patterns – authors refer to a chapter, which I specifically looked for and apparently it explains nicely what structural defects exist in the crystal and provides 1 example of the effect of stacking faults on powder XRD – which is irrelevant to current study. Therefore, instead of this reference author should cite something more relevant. For example, there is a “bible” of analysis of structural defects by TEM written by JW Edington “Practical Electron Microscopy in materials science” (1976). This book also has wide explanation of satellites reflections and double diffraction phenomenon – which should be reviewed by authors, since I believe that double diffraction is not the effect seen on their patterns.

Answer: Thank you for your advices. We have replaced the reference of “David B. Williams, C. Barry Carter. Transmission Electron Microscopy: A Textbook for Materials Science (Springer, 1996)” with “Edington, Jeffrey William. Practical Electron Microscopy in Materials Science (Philips Technical Library, 1976)”.

By carefully checking the SAED pattern we obtained, the phenomenon we observed should be the superposition of the rotation of two electron diffraction patterns in L-CTD sample. We have added a new Figure in Supplementary Information as Figure 29, revealing the superposition of two sets of ED patterns with 4.1° rotational angle. The corresponding content of the manuscript has been also revised at the bottom of the left column on page 5.

Figure R2. HRTEM image and the corresponding SAED pattern taken from [001] zone axis of the plate-like particle in L-CTD. **a**, The TEM image of overlapped two nanosheets. **b**, The corresponding SAED pattern. The average central angle (4.1°) between the visible diffraction spots to the origin represents the rotational angle between the two nanosheets.

2) In Table 1 in extended data and throughout all article authors provide lattice parameters with 3 digits after the dot. Is 0.001Å really the accuracy of the determination of lattice parameters using EDT? If you state such an accuracy – please look at your own results, where you wrote that for L-CNM from datasets/crystals 1 and 2 you got $a=9.535\text{Å}$ and $a=9.941\text{Å}$ (for example). If you consider them as being the same, the accuracy is 0.4Å. So, why you put everywhere 3 digits after the dot? Same goes about the two digits after the dot in the angles.

Answer: Thank you for your suggestion. For the simplest case with only 2 measurements, the correct way of error estimation would be to provide 2 values: (i) the mean absolute deviation that is $[|a_1 - a_{\text{avg}}| + |a_2 - a_{\text{avg}}|]/2 = 0.2 \text{ Å}$ (where $a_{\text{avg}} = (a_1 + a_2)/2 = 9.738 \text{ Å}$) and (ii) the standard deviation that is $\text{stdev} = \text{SQRT}([|a_1 - a_{\text{avg}}|^2 + |a_2 - a_{\text{avg}}|^2]/(N-1)) = 0.29 \text{ Å}$. For the best experience the statistics should be based on a larger number of data sets. The unit cell parameters and angles determined by software are usually compared to the standard XRD JCPDS card due to the well-known precision measurement issues in TEM raw data sets. (Sun, Q. *et al.*, *J. Mater. Chem. A*, **2**, 17828–17839 (2014); Gemmi, M. *et al.* *Z. Kristallogr.* **228**, 51-58 (2013); Kolb, U. *et al.*, *Cryst. Res. Technol.* **46**, 542–554 (2011)). For ensuring the accuracy of the results, we have modified the cell parameters by keeping one digit after the decimal point (Table R1). For D-CNM (A_1 , the top area of the rod-like particle), the unit cell parameters determined from 3D EDT software show a maximum error of 3.5% ($\sim 0.3 \text{ Å}$) and 0.5° as the maximum deviation from the ideal β angle of 114.22° compared with the standard JCPDS card. The maximum value of mean absolute deviation and the standard deviation of the unit cell parameters calculated from all 3D ED data are 0.14 Å and 0.17 Å , respectively. Similarly, the maximum error in unit cell parameters of L-CTD determined by 3D EDT is 3.4% ($\sim 0.2 \text{ Å}$), and the maximum value of mean absolute deviation and the maximum standard deviation

from all 3D ED data are 0.08 Å and 0.10 Å, respectively. However, the error can be reduced using the averaged unit cell parameters calculated from several datasets, which changed to 1.7% for CNM and 1.2% for CTD, suggesting the error can be effectively reduced to 2%. The corresponding discussions have been added in the main text and the Supplementary Information Table 1 and Table 5, respectively.

Table R1. Lattice parameters of CNM

	a	b	c	β	Space group
D-CNM (A ₁)	9.9 Å	8.8 Å	7.8 Å	114.7°	C2/m (12)
D-CNM (A ₂)	9.6 Å	8.9 Å	7.7 Å	114.2°	C2/m (12)
D-CNM ₁	9.9 Å	8.7 Å	8.0 Å	114.4°	C2/m (12)
L-CNM (B ₁)	9.6 Å	8.9 Å	7.7 Å	114.2°	C2/m (12)
L-CNM (B ₂)	9.7 Å	8.9 Å	7.7 Å	114.3°	C2/m (12)
L-CNM ₁	9.5 Å	8.6 Å	7.6 Å	114.2°	C2/m (12)
L-CNM ₂	9.9 Å	8.8 Å	7.8 Å	114.8°	C2/m (12)

Table R2. Lattice parameters of L-CTD

	a	b	c	Space group
L-CTD	4.8 Å	5.9 Å	16.1 Å	Pbcn
L-CTD ₁	4.8 Å	5.8 Å	16.0 Å	Pbcn
L-CTD ₂	4.8 Å	5.7 Å	16.0 Å	Pbcn
L-CTD ₃	4.7 Å	5.7 Å	16.0 Å	Pbcn

3) The general question of accuracy/tolerance/error of measurements/method is left unanswered. In my opinion, one cannot suggest a method without evaluation of its precision. What the accuracy in determination of the twist angle, pitch length etc` is? It is of high importance since the assessment of these values is the major discovery of this article. You state in most placement “approximately”. In my view some small discussion and evaluation of the precision should be made. My guess will be that the angle determination precision is not better than 1.5-2 degrees.

Answer: Thank you for your suggestion. In this paper, the precision is mainly determined by the precision of the TEM goniometer tilt axis. The goniometer tilt angle accuracy is mechanically controlled by a stage motor. To reduce the mechanical backlash, the goniometer should be tilted consequently in the same direction. The errors in the alpha-tilt of the goniometer in TEMs can be $\pm 1^\circ$. In the experiments employing beam tilt, it guarantees that we have a high beam tilt precision within 2-3° because it is electronical. After collection 3° using the beam tilt, we do a mechanical tilt, and we drop back to the mechanical precision. For the continuous collection method, the goniometer is tilted constantly in the same direction. Once the motor in the goniometer accelerates to a constant speed after initial start, its error also stabilizes. For more general cases, an error of $\pm 1.5^\circ$ should be reasonable estimation.

Besides, the precision can be also affected by the angle measurement error, which depends on the precision of diffraction spot positions against the central beam position. The individual diffraction spot centers can generally be calculated with a high precision, which mainly depends on their resolution and the diffraction intensity distribution in the electron diffraction pattern: the further and sharper the spots, the higher the precision. The central beam position calculation requires special attention since all ED spots will be calculated relative to it. The primary beam coordinates can be

refined from the available Friedel pairs. Again, statistically speaking, the more Friedel pairs an ED has the higher the precision of the central beam position. In conclusion, our experience shows that the maximum error for measurement can be less than 0.5° .

To achieve the best precision in the ED data measurements, we could use the internal standard for the calibration of each electron diffraction pattern in the 3D ED data set to compensate for the sample height variation and the TEM projector lens distortion. Simultaneously, for the precise measurement of the alpha-tilt angle, we could use external electronics for precise angle tracking. However, all the improvements would require significant modifications to the data collection and processing software packages.

We have added these contents in the Discussion part in the manuscript.

4) Beam stopper was used during the acquisition of all datasets. To the best of my knowledge, usage of a beam stopper, makes it more difficult to merge the frames. This might lead to distorted reflections, sometimes even reflections which look a little bit elongated and bent. Here, appearance of the reflections is of an utmost importance. Please refer to this problem since I am speaking out of experience of having hard time merging the data with the beam stopper.

Answer: The EDT-PROCESS software that we use for the 3D ED data processing was initially designed to work with beam tilt experiments and can be also used for the datasets only employing goniometer tilting. Each subset of 3° of ED frames acquired using the TEM beam tilt is aligned using cross-correlation and summed into a single frame. Since each step is relatively small, the consequent ED patterns variation is relatively small and not so many spots vanish or appear. This allows the processing software to successfully use cross-correlation to compensate for the electron beam drift. We can check each summed frame and visually control the correctness of the beam drift estimation done by the software. In rare cases of spot elongation that can be easily detected visually, various options for controlling the frames alignment algorithm can be used, which makes it easy to correct the beam drift compensation.

In the case of twisting, the spot elongation appears systematically as circular shapes around the common center, while a wrong frame alignment shifts all spots in the direction of the beam drift. These two cases can be easily distinguished. Simultaneously, we can note that every spot elongation due to the wrong alignment leads to the same spot shapes, e.g., lines, and always can be corrected. However, twisting cannot be corrected during the 3° beam tilt drift compensation procedure because the spots are placed according to the twisting rotation which is angular feature and not linear.

In the case of the alignment of 3° sub-sets in 3D, we can use the calculated sub-set centers (see above, Friedel pairs). Spot elongation in the forms of arcs can appear in this scenario only when: (i) the goniometer alpha-tilt axis is calculated wrong, (ii) the pixel size is wrong, and (iii) the direction of the beam tilt is wrong. All these parameters can be modified and their effect on the data visualization can be checked in the software. In all cases of the wrong alignment, the software allows to fit the correct parameters and completely compensates for the distortion or elongation, except for the diffuse scattering. However, for twisted structures, such a correction becomes impossible. We checked all 3D datasets for twisted structures, and couldn't find any combination of alpha- and beam-tilt angles to fully compensate for elongation/bending: either side of 3D reciprocal space gets worse as the other side gets closer to "ideal".

5) Methodology: not much information is given about the process of acquiring the EDT data. Was it continuously? Was it step-wise? If step wise – what was the step value? Was it in STEM mode? In TEM? Using nanobeam? Using the selected area aperture? Size of the aperture?

Answer: We have added the experimental details in the Methods section. The CTD samples were studied and verified by step-wise data collection based on a combined stage-tilt/beam-tilt collection scheme in a JEOL JEM-2100 transmission electron microscope (point resolution 0.25 nm, lattice resolution 0.14 nm, Cs = 1.4 mm, Cc = 1.8 mm) using the 3rd SA aperture in the SAED mode with ~200 nm in diameter. The 3D ED data was collected using a combination of goniometer tilt with a step of 3° (with 0.2° overlap) combined with electron beam tilt with a rotation step of 0.1°. The CNM samples were investigated by a recently developed continuous automated data acquisition based on the goniometer-tilting scheme in a JEOL JEM-2100 Plus transmission electron microscope (point resolution 0.23 nm, lattice resolution 0.14 nm, Cs = 1.0 mm, Cc = 1.4 mm), which reduced the acquisition time from approximately one hour to seven minutes. The 2nd SA aperture with 700 nm in diameter was chosen. The camera length is 600 mm and the individual frame exposure time is 500 ms. 3D ED data collected by both methods show consistent torsion axis and arched-shape diffraction intensity distribution modulations.

6) While working in the TEM and taking the EDT datasets – the crystal heats up. So, there are effects of twisting that you are looking for, but there are also bending of the crystal just because it is on the carbon film and not planar, in addition, it bends during the TEM analysis due to heating. These effects were ignored by the authors. Please elaborate.

Answer: Thank you for your comments. In our observations, the samples were prepared after scraping off the substrate followed by slicing the sample embedded in epoxy resin, which had better stability compared with the ordinary sampling using carbon film. When taking the diffraction pattern, the low dose condition was applied to prevent sample damage. Upon manually identification process, after the SAED patterns were taken by aligning the zone axis from different regions of the twisted crystal to the electron beam, we go back to the original position and the tilting angle of the first region to check if the sample is unchanged. Similarly, after 3D ED data collection, we also go back to the initial collection angle of the alpha-tilting axis to ensure the stability of the sample. As shown in the Figures R3 and R4, no obvious change in crystal shape of the two samples were observed during one hour of data acquisition using the step-wise EDT method, indicating that the crystal has good thermal stability. Therefore, the electron beam irradiation has little effect on the deformation of crystal structures in our TEM analysis. We have added the corresponding discussions and the crystal shape changes in the process of collecting 3D ED data of L-CTD sample in Supplementary Fig. 20.

Figure R3. The images automatically taken by the EDT-COLLECT software of the same D-CNM particle during the data collection process. Each image is taken for every goniometer tilting axis of 3° . During an hour of automatic acquisition, no damage or deformation of the crystal can be observed. The particle remains intact after 3D ED data collection and the ED pattern can be reproduced by going back to the original angle, indicating the crystal has good thermal stability.

Figure R4. The images automatically taken by the EDT-COLLECT software of the same L-CTD particle during the data collection process. Each image is taken for every goniometer tilting axis of 3° . During an hour of automatic acquisition, no damage or deformation of the crystal can be observed. The particle remains intact after 3D ED data collection and the ED pattern can be reproduced by going back to the original angle, indicating the crystal has good thermal stability.

b) Specific points:

1) Page 3, left column, bottom lines— estimation of the space group of the crystal d-CNM. According to the stated extinction conditions, the only conclusion which can be made is that the extinction (diffraction) symbol is C1-1, meaning that 3 space groups can be assigned to this data: C2 (number 5) (full symbol C121); Cm (number 8) (full symbol C1m1) and C2/m (number 12) (full symbol C12/m1). This information can be found in the International Tables of Crystallography volume A, which, by the way, was not cited and should be cited both here and in the second place, where the space group Pbcn was estimated (lucky for authors the Pbcn space group is indeed the only one related to the Pbcn extinction symbol 😊). So, summarizing my remarks: a) cite international tables in 2 places; b) show that there are in fact three options which comply with obtained extinction conditions; c) explain why you have chosen C2/m out of these three options. Even if it was not in scope of your research
– it must be done correctly.

Answer: Thank you for your advice. We have supplemented the text with the reference (ref 41) of the International Table in the two places. In determining the crystal structure of nickel molybdate, we agree that C2/m cannot be unique chosen because there are three possible space groups that meet the extinction conditions. The space group was chosen with the help of the Open source Materials Project Database (<https://materialsproject.org/>) and Inorganic Crystal Structure Database (<https://icsd.fiz-karlsruhe.de/search/>) with the following possible structural forms: NiMoO₄ (P2/c), NiMoO₄ (C2/m), Ni₂Mo₃O₈ (P6₃mc), NiMo₄O₁₅ (P1), Ni₄Mo₄O₁₉ (P1), NiMoO₄ (C2/c) and NiMoO₃ (Pm3m). The only possible configuration of nickel molybdate crystal structure is NiMoO₄ with space group C2/m. Wide-angle X-ray diffraction also confirmed the reflections in the resulting spectrum are in accordance with Ni(MoO₄) (JCPDS card No. 86-0361) with the space group C2/m. Therefore, we have modified the sentence to “The reflection conditions deduced from the 3D reciprocal lattice meet the systematic extinction rules⁴¹ of *hkl*, *hk0*, *0kl*: $h + k = 2n$; *h0l*, *h00*, *00l*: $h = 2n$ and $0k0$: $k = 2n$, suggesting the three possible space groups C2 (No. 5), Cm (No. 8) and C2/m (No. 12), however, the only possible space group C2/m (No. 12) is chosen by comparing all possible space groups of molybdenum nickel oxide” at the bottom lines of the left column on page 3 in the manuscript.

2) Page 4 bottom line in the left column vs midsection in the right column: I would expect to find in these two places the same lattice parameters, but they are different. Is the right one from the EDT and the left one from the JCPDF card?

Answer: The unit cell parameters shown on page 4 bottom line in the left column (top right column on page 4 of revised manuscript) were determined by wide-angle X-ray diffraction and it was in line with the standard XRD JCPDS card No. 78-1063 with the space group *Pbcn* (No. 60) and lattice parameters of $a = 4.737 \text{ \AA}$, $b = 5.708 \text{ \AA}$ and $c = 15.865 \text{ \AA}$, and the unit cell parameters of $a = 4.8 \text{ \AA}$, $b = 5.9 \text{ \AA}$ and $c = 16.1 \text{ \AA}$ shown on page 4 midsection in the right column is determined by the 3D EDT method. To clarify, we have revised the sentence to “The sample was determined to be mainly composed of the *Pm*-cassiterite phase [WWW-MINCRYST, OXIDE_Sn-3381, JCPDS card. No. 78-1063] with the space group *Pbcn* (No. 60) and lattice parameters of $a = 4.737 \text{ \AA}$, $b = 5.708 \text{ \AA}$

and $c = 15.865 \text{ \AA}^3$, and the fine crystal structure could be determined via the 3D EDT method (*vide post*)”.

3) Page 4, right column, line 15 from the top – the precise determination of the angle deviation (such as 0.49 degrees, 0.55 etc`, stated in the extended data) is far beyond the capability of TEM, to the best of my understanding. I don't know which size of the selected area aperture was used there (extended data fig 19 a), but assuming you draw the exact size of the aperture, you have many overlapping position-wise. Your pattern contains averaged reflections` intensities originated from the same position with additions. In some places more inclination, since you are taking less of the “overlapped place”. Going back to the accuracy – 0.49 degrees... Please explain since analysis of intensity here does not apply! Furthermore, in any crystal that I work with (which are not chiral and not twisted) I see this effect due to natural crystal/sample bending.

Answer: Thank you for your comments. This diagram illustrates lattice torsion in terms of the intensity and angular rotation of the diffraction spots in the SAED patterns of the particle. Because the torsion angle is very tiny within the small size of crystal, it is difficult to manually tilt the particle to align well with the electron beam. Therefore, the SAED patterns were taken from several contiguous positions of one typical plate. Although this method is not precise enough, a rotational relationship with the angle deflection indicates a possible distortion of crystal lattices into the crystal bending or twisting arrangement. We admit that the overlapped parts make the information from same position with additions and the interpretation is very complicated. In the revised manuscript, we removed the overlapped part and kept the SAED patterns taken from the three adjacent non-overlapping regions (Figure R5 b-d). The slight rotation of the SAED pattern can be well identified. The angles measured here are also revised for one significant digit. We also agree that this kind of phenomenon can be caused by natural crystal bending. However, it is impossible to distinguish from the conventional SAED data. The true nature of the crystal can be revealed by the 3D EDT method. We have revised the manuscript to clarify this point. The corresponding deflection angle has been modified on the top area of right column on the page 4 of the manuscript and the Figure R5 has been replaced in Supplementary Information Fig. 19.

Figure R5. TEM image and its corresponding SAED patterns taken from $[01\bar{1}]$ zone axis of the plate-like particle in L-CTD. **a**, TEM image of the plate-like particle in L-CTD and the black circles represent the three contiguous positions of SA aperture for taking the SAED patterns. The inset shows the schematic drawing of the plate-like particle. **b-d**, The corresponding SAED patterns. By the SAED patterns taken from three contiguous positions of the plate-like particle with the side length of ~ 270 nm, a rotational relationship with a tiny deflection of 1.1° was observed along the $[01\bar{1}]$ zone axis in a clockwise manner. Furthermore, the diffused intensity and split diffraction spots can be observed for the high index reflections (the red and yellow arrows of **d**), indicating a possible distortion of crystal lattices into the crystal bending or twisting arrangement.

4) Page 4 bottom part of the right column – I strongly disagree with the authors. I recommend them to check the book of Edington, which I referred to earlier in my report. Dislocations – will not produce any effect on the ED pattern. So, they are irrelevant to the change in the appearance of reflections. Second – from this one HRTEM image – we cannot conclude what exactly we see here, especially since we are looking into crystal, which is twisted due to chirality, as you state. May be these effects, which you point to in fig 26 in extended data, regard to off-zone axis conditions? Or local bend/strain? Or maybe local thickness variations? Stacking faults are seen nicely when they are “edge on” at specific orientation at bright field and/or dark field low magnification TEM images, see book written by Edington. Furthermore, the reflections will not move due to structural defects, they might provide satellites in specific cases. Please revise this part.

Answer: Thank you very much for your advice, which is very helpful to us. We have referred the book by Edington and revised the manuscript. In our opinion, the position change of diffraction spots may be due to the excitation error (the deviation from the Bragg reflection position) owing to

the off-zone axis condition due to the shape effect of specimen twisting, the distortion of the reflecting plane into the Bragg position caused by rapidly varying strain field near the dislocation core, the change of intensity distribution induced by local thickness variations and the measurement deviation from the exact Bragg position on the grounds of the curvature of the Ewald sphere. We have added these discussions in the middle part of the left column on page 5 in the manuscript.

5) Page 5 – reference to double diffraction phenomena. Will the double diffraction spots move? It seems more like additions, originated by the layer beneath/above the original crystal which is twisted/shifted (i.e., not at the exact zone axis conditions) and, therefore, peripheral additional reflections shift as a function of misalignment with the upper/lower layer/crystal. Most likely the patterns are dynamical, but these reflections are not necessarily originated by double diffraction. I could not find which original reflection duplicates to reproduce itself at the position in the pattern which you mark. I want to emphasize that I do not argue with your results – the satellites due to twist/shift are clearly seen – I argue with the reason of their appearance.

Answer: Thank you for pointing out. By carefully checking the SAED pattern we obtained, the observed phenomenon can be attributed to the superposition of rotated diffraction patterns derived from the respective contributions of the layered rotational nanosheets. Due to the rotationally stacked arrangement of the upper and lower crystals, the higher-order diffraction spots are more dispersed and the lower-order diffraction spots appear to overlap. The corresponding content has been amended to the bottom on the left column of page 5 in the manuscript.

Reviewer #3:

The paper presents important news of the structural characterizations of chiral mesostructured inorganic assemblies, which is particularly challenging and of great importance. The highlight of this paper is the full analysis of reciprocal and real space data that enables the determination of the mesostructure that is almost impossible to be solved by conventional techniques. The TEM analysis is state-of-the-art and the discussion related to chiral arrangement is very solid. From my point of view this work has good novelty and is groundbreaking, suitable for publication in Nature Communications after some minor revisions.

1. Relatively little information was provided for the materials employed in this investigation. The rationale for the synthesis of the chiral mesostructures can be included in the experimental section or in the supplementary information.

Answer: Thank you for your advice. The chiral mesostructured inorganic crystals were normally formed by introducing the asymmetry units or the asymmetric interaction on the surface and its environment. Particularly, the two examples in our manuscript were synthesized by a hydrothermal process using chiral amino acid molecules as the symmetry-breaking agent and structure-directing

agent, and the inorganic source as precursor. Synthesis strategies of the materials have been added in the Methods section on the left column of page 7.

2. Add scale bar for the electron diffractions shown in Fig. 2 and Fig. 3.

Answer: Thank you for your suggestions. The text diagrams have been changed accordingly.

3. In Fig. 2 and Fig. 3, “The amplitude of the diffraction is set to be square root for visualization.” Please clarify whether the display mode of the data affects the position judgment of the diffraction intensity.

Answer: The position of the spots is not affected by the variation of the intensity because we only modify the visual presentation of the intensities and not their positions. By taking this configuration, the weak diffraction spots can be realized otherwise it may be hard to see the details. The visualization does not affect the overall analysis because the analysis is done based on the measured intensities. To clarify, the sentence was revised to “The diffraction spots grayscale intensities are square roots of the corresponding measured values for the better visual experience during the publication.”

4. In the conclusion section, the authors should state the types of materials this method is able to resolve.

Answer: Thank you for your advice. In the end of the article, we have clarified that the 3D EDT technique can not only be used to determine the multilevel chirality of chiral mesostructured inorganic materials which process distinctive twisting and helical hierarchical stacking, but also be applied to mesostructured crystals with bending, coiling and other deformations.

Reviewer #4 (Remarks to the Author):

Overall this is a nice paper. I am not an expert in the area of Chiral crystals, so I can only look at the work in a general sense. I see nothing technically wrong with the work. However, the authors fail to make any case for it to appear in NatComm; as written it is much more appropriate for Acta Cryst or Ultramicroscopy.

Answer: Thank you for your approval on the quality of our manuscript. The chiral inorganic mesostructured material is one of the most captivating fields in nanotechnology and chiral natural sciences (Yu, S. *et al.*, *Nat. Mater.* **4**, 51-55 (2005); Jiang, W. *et al.*, *Nat. Comm.* **8**, 15066 (2017); Lee, H. *et al.*, *Nature.* **556**, 360-365 (2018); Jiang, W. *et al.*, *Science.* **368**, 642-648 (2020)). It spurs the development of characteristic structural-chiral anisotropy properties such as chiroptical activity (Ma, W. *et al.* *Chem. Rev.* **117**, 8041-8093 (2017)), photomagnetic-chiral anisotropy (Liu, Z. *et al.* *Chem.* **8**, 1-11 (2021)), enantiospecific discrimination and catalysis (Liu, Z. *et al.* *Angew. Chem. Int. Ed.* **59**, 15226-15231 (2020); Li, S. *et al.* *Nat. Comm.* **10**, 4826 (2019)), enantiomer-dependent

immunological response (Xu, L. *et al. Nature*. **601**, 366-373 (2022)), etc. Synchronous analysis of these unique multilevel chirality of chiral inorganic mesostructured materials is significant for investigating their chemical and physical properties. In this way, we believe this topic will attract a lot of attention from readers. In the revised manuscript, we have modified the introduction part to include more aspects of materials science. We hope this revision will be suitable for more general audiences.

Main comments, including more on the scope of the paper are:

1) It took me a few moments before I understood the focus of the paper, for which Fig. 1 is very useful. I think the first paragraph which start "In contrast.." needs to refer to Fig. 1 mentioning the different types of crystals present. This will significantly help the reader, who otherwise has to wait until the second page.

Answer: Thank you for your advice. We have referred to Fig. 1 in the first paragraph when describing the chiral mesostructured inorganic crystals.

2) I am horrified by the last sentence of the paper:

"This project may promote the study of electron crystallography and provide significant opportunities in developing new chiral materials."

There is nothing of sublime importance in electron crystallography, nobody will get a Nobel for it by itself. What matters is the science that it can reveal.

Answer: Thank you for your comments. We are sorry for the improper description here. The last sentence of the paper has been changed to "This project may promote the application of 3D electron crystallography in the structural elucidation of different kinds of materials, and provide new ideas for fundamentally understanding the structure-activity relationship of materials and the designing of new functional materials."

3) Related to point 2), as it stands this paper should appear in either *Acta Cryst* or *Ultramicroscopy*, as it is all technique. The authors have to make the case for the larger science. For instance, are their chiral materials used in biology or applications where the macro-structure is unknown? I suspect that they can, they will have to in a revised version.

Answer: Thank you for your comments and suggestions. To make the article more universal and suitable for general audiences, we have revised the introduction of the manuscript as follows "Chiral mesostructured inorganic materials were widely reported to process the characteristic structural-chiral anisotropy properties, such as chiroptical activity, photomagnetic-chiral anisotropy, enantiospecific discrimination and catalysis, enantiomer-dependent immunological response. Unlike chiral crystals with chiral space groups, chiral inorganic mesostructures can be formed by twisting of crystal structure with achiral space groups or stacking of primary assembly units into a hierarchical arrangement."

The structural-chiral anisotropy properties of chiral mesostructured inorganic materials are suitable for a variety of physical, chemical, and biomedical applications. The two samples in our manuscript show the mirror-imaged optical activity signals, revealing the photochiral selectivity of the chiral

mesostructures. The optical activities can be also discovered in chiral TiO₂ nanofibres, chiral CuO mesostructured nanoflowers, chiral mesostructured BiOBr films, etc. (Liu, S. *et al. Nat. Comm.* **3**, 1215 (2012); Duan, Y. *et al. J. Am. Chem. Soc.* **136**, 7193-7196 (2014); Ding, K. *et al. Angew. Chem. Int. Ed.* **60**, 19024-19029 (2021)). Moreover, chiral nanostructured gold films and chiral mesostructured NiO films have been also reported to process the photomagnetic-chiral anisotropy (Liu, Z. *et al. Chem.* **8**, 1-11 (2021); Bai, T. *et al. Angew. Chem. Int. Ed.* **60**, 9421-9426 (2021)), and chiral mesostructured half-metallic Fe₃O₄ films own the resistance-chiral anisotropy (Bai, T. *et al. Angew. Chem. Int. Ed.* **60**, 20036-20041 (2021)). Regarding the biological related applications, the enantioselective interaction has been recently found to exist between cells and chiral hydroxyapatite films (Zhou, C. *et al. Chem. Mater.* **34**, 53-62 (2022)) and the enantiomer-dependent immunological response to chiral nanoparticles (Xu, L. *et al. Nature.* **601**, 366-373 (2022)). Currently, this paper focuses on the structural solutions of chiral materials. In the future, it is expected that new chiral structures and assembly modes will be thoroughly investigated by this method and their structure-activity relationships will be revealed. After revision of the introduction and the conclusions, we hope that this manuscript is more suitable for general audience.

REVIEWER COMMENTS

Reviewer #1 (Remarks to the Author):

This referee is satisfied with the authors reply and consider the paper ready for publication.

Reviewer #2 (Remarks to the Author):

Authors have answered all my questions, besides the matter of dislocation core effect. They insist on dislocation as a reason for the effect but, in fact, using selected aperture of 200-700nm in size, as stated by authors, I wonder if the effect of strain near the dislocation core will be seen on the ED data. I cannot accept that without seeing some references as an example of effect of dislocations effect on ED data, while using SAED mode (and not nanobeam).

Reviewer #3 (Remarks to the Author):

The authors have responded all my concerns. I have no more questions at this stage and recommend it published in its current revision.

Reviewer #4 (Remarks to the Author):

I am going to leave the decision on this paper to the editors. In my opinion the authors have not tried to answer my criticism. The first paragraph that they have added completely fails to justify the work in the broader context; as I said before, what is the big picture science of this work?

For the last sentence, all that they have done is rephrase the point about 3D electron crystallography.

My opinion remains that this work belongs in Ultramicroscopy or Acta Cryst, as there is no real justification for this in the broader context, either in the original or the revised. The only people who will be interested in this work, as written, is the very small community who work with electron diffraction; even then, it will be a small part of that community!

Point-by-point response to the reviewers' comments

Reviewer #1 (Remarks to the Author):

This referee is satisfied with the authors reply and consider the paper ready for publication.

Answer:

Thank you very much for your suggestion.

Reviewer #2 (Remarks to the Author):

Authors have answered all my questions, besides the matter of dislocation core effect. They insist on dislocation as a reason for the effect but, in fact, using selected aperture of 200-700nm in size, as stated by authors, I wonder if the effect of strain near the dislocation core will be seen on the ED data.

I cannot accept that without seeing some references as an example of effect of dislocations effect on ED data, while using SAED mode (and not nanobeam).

Answer:

Thank you very much for your suggestion. We are sorry for misinterpreting the dislocation effect in the book by Edington. The dislocation should only influence the shape of diffraction spots without causing the deviation in the Bragg reflection position. We checked the bright field HRTEM image and the corresponding SAED pattern using the 4th SAED aperture (~100 nm in diameter) from the edge part of the plate-like L-CTD sample. Although the diffusion streaks can be recognized in the SAED pattern, the diffraction spots are located on reciprocal lattice sites as indicated by the red reference lines. We agree with the reviewer that the dislocation cannot be revealed using SAED mode.

Therefore, we deleted the improper descriptions and revised this content to “Notably, not all the diffraction spots are located exactly on the reciprocal lattice site, which may be the excitation error (the deviation from the Bragg reflection position) owing to the off-zone axis condition due to the shape effect of specimen twisting or the change of intensity distribution induced by local thickness variations and the measurement deviation from the exact Bragg position on the grounds of the curvature of the Ewald sphere.” in the middle part of the left column on page 5 in the manuscript.

Figure R1. TEM image and its corresponding SAED pattern taken from $[1\bar{1}0]$ axis of the plate-like particle in L-CTD. a, TEM image of the plate-like particle in L-CTD. **b,** The corresponding SAED pattern of the particle using the 4th selected aperture in the SAED mode with ~ 100 nm in diameter. The typical diffraction spots are marked by blue circles. The dotted red lines are the reference lines for the location of the diffraction spots in the SAED pattern.

Reviewer #3 (Remarks to the Author):

The authors have responded all my concerns. I have no more questions at this stage and recommend it published in its current revision.

Answer:

Thank you very much for your suggestion.

Reviewer #4 (Remarks to the Author):

I am going to leave the decision on this paper to the editors. In my opinion the authors have not tried to answer my criticism. The first paragraph that they have added completely fails to justify the work in the broader context; as I said before, what is the big picture science of this work?

For the last sentence, all that they have done is rephrase the point about 3D electron crystallography.

My opinion remains that this work belongs in Ultramicroscopy or Acta Cryst, as there is no real justification for this in the broader context, either in the original or the revised. The only people who will be interested in this work, as written, is the very small community who work with electron diffraction; even then, it will be a small part of that community!

Answer:

Thank you for your comments. We regret that we were unsuccessful in conveying our scientific messages. We sincerely want to clarify that this manuscript is not only suitable for people working on electron diffraction and crystallography, but also for chemists and materials scientists working on the related materials.

Chirality is an important concept for stereochemistry. The study of chiral materials is of great significance to promote the development of chiral catalysis, chiral separation, optoelectronics, biomedicine, etc. With the development of nanotechnology and supramolecular chemistry, more attention is paid to self-assembly of multilevel functionalized chiral mesostructures. These chiral mesostructured inorganic materials may have twisting nanostructures or helically stacking of crystal lattices, leading to extraordinary materials properties. The structural characteristics of hierarchical chirality also distinguishes these materials from previously studied chiral structures composed of chiral space groups. The number of publications addressing this topic increased rapidly in the past decade.

Therefore, the in-depth structural analysis and the determination of hierarchical chirality of these materials are not only the key to understanding their properties, but also facilitate the development of structural-chiral anisotropy in related physical, chemical, and biological applications. However, the synchronous structural characterizations of these spiral arrangement assemblies, especially distinctive twisting and helical hierarchical stacking chirality in chiral materials synthesized by achiral space groups, remain many bottlenecks due to their geometrical complexity and the restrictions on the current techniques. Therefore, we focused on this topic and proposed the new approach based on three-dimensional electron diffraction tomography (3D EDT).

The paper is also closely related to synthesis and structural solution of mesostructured materials, which were widely discovered in many natural and artificial systems. Our method provides a new idea for solving their complex hierarchical arrangements. Besides, although 3D EDT has been employed to solve several zeolites, metal–organic frameworks and covalent organic frameworks, it has never been applied to reveal multilevel chirality. The advancement of the 3D EDT will also draw the attention of people in the field of structural analysis.

We believe that the proposed method for the determination of the multilevel chirality of chiral mesostructured inorganic crystals will attract broad audience, particularly who are interested in chiral synthesis, mesostructured characterization technology, and characteristic applications.

To make the paper more attractive to general audience, we further revised the introduction part as follows “Chirality is an essential characteristic of nature, and it is ubiquitous from the microscopic to the macroscopic scales. The study of chiral materials is of great significance to promote the development of chiral catalysis¹, chiral separation^{2,3}, optoelectronics⁴, biomedicine⁵, etc. With the development of nanotechnology and supramolecular chemistry, more attention is paid to self-assembly of multilevel functionalized chiral mesostructures. Particularly, chiral mesostructured inorganic materials have attracted great attention due to their characteristic structural-chiral anisotropy properties, such as chiroptical activity^{3,6}, photomagnetic-chiral anisotropy⁷, enantiospecific discrimination and catalysis^{8,9}, enantiomer-dependent immunological response¹⁰, etc. Unlike chiral crystals with chiral space groups, chiral inorganic mesostructures can be formed by twisting of crystal structure with achiral space groups or stacking of primary assembly units into a hierarchical arrangement¹¹⁻¹⁴. The structural characteristics of hierarchical chirality also

distinguish these materials from previously studied chiral structures composed of chiral space groups. Three typical types of chiral mesostructured inorganic crystals are shown in Figs. 1c₁-1e₁. Understanding the unique multilevel chirality of materials is a key starting point for investigating their chemical and physical properties as well as the synthesis conditions. It is therefore extremely important to develop new structural characterization techniques for the determination of hierarchical chirality in these materials....”

We also revised the abstract part as “Chiral mesostructured inorganic material is one of the most attractive fields in nanoscale materials presently. It spurs the development of novel structural characterization techniques and characteristic structural-chiral anisotropy properties. The chiral mesostructures exhibit distinctive twisting and helical hierarchical stacking ranging from atomic to micrometre scales. A detailed analysis of these materials is key to understanding their characteristic chiral anisotropy....”

We also added the structure-property relationship content as “It can be concluded that the primary chirality with twisted crystal lattices in CNM is the origin of the significant chirality-dependent OA response.” in the bottom part of the left column on page 4 in the manuscript. Besides, we added the corresponding content of “It is the hierarchical chirality in the mesostructure of CTD that contributes decisively to OA signalling. These results clearly reveal the structure-activity relationship of the chiral mesostructured materials.” to the middle part of the left column on page 6.

For the last sentence, we made further revision as follows “This project will play an important role in the structural elucidation of different chiral mesostructured inorganic materials and other mesostructured materials in the future. It may also promote the development of new characterization methods for crystal defects and arrangements, fundamentally enhance the understanding of the structure-activity relationship of new materials, and facilitate the fabrication of new functional materials with spatial geometric variation.”

We sincerely hope this revision will be suitable for more general audiences.

REVIEWERS' COMMENTS

Reviewer #2 (Remarks to the Author):

The authors have responded all my concerns. I have no more questions at this stage and recommend it published in its current revision.

Reviewer #4 (Remarks to the Author):

The authors have not added anything that convinces me that this is of general interest. In fact, their latest revision made the article significantly worse, so my recommendation is now reject. The authors need to learn some higher-level electron diffraction, they are making beginners mistakes.

The statement they make "The dislocation should only influence the shape of diffraction spots without causing the deviation in the Bragg reflection position." is completely wrong, and indicates that they do not know basic electron diffraction. The local strain field can be considered as a change in the excitation error, and also causes a shift -- just write down the phase as a function of position and take the derivative. This has been known since the earliest images of artifacts due to varying thicknesses being misinterpreted as dislocations more than 50 years ago!

Furthermore the paragraph starting "Notably, not all..." that they added in response to referee #2 is horribly wrong. The authors seem to believe that diffraction spots change position with the excitation error. This is completely wrong, it depends upon the combination of the excitation error and the surface normals. No paper should be published with this text, it will move science backwards. While the book by Eddington is useful, it is not a high level text. The authors need to read and understand topics such as diffractive multiplets as discussed in Cowley's book; they should also read and understand the book by Peng, Dudarev and Whelan is a more modern text, also probably harder to understand as it goes deeper.

My opinion is now a definitive reject, and send to Acta Cryst.

Point-by-point response to the reviewers' comments

Reviewer #2 (Remarks to the Author):

The authors have responded all my concerns. I have no more questions at this stage and recommend it published in its current revision.

Answer: The authors thank the reviewer #2 for the evaluation of the manuscript.

Reviewer #4 (Remarks to the Author):

The authors have not added anything that convinces me that this is of general interest. In fact, their latest revision made the article significantly worse, so my recommendation is now reject. The authors need to learn some higher-level electron diffraction, they are making beginners mistakes.

The statement they make "The dislocation should only influence the shape of diffraction spots without causing the deviation in the Bragg reflection position." is completely wrong, and indicates that they do not know basic electron diffraction. The local strain field can be considered as a change in the excitation error, and also causes a shift -- just write down the phase as a function of position and take the derivative. This has been known since the earliest images of artifacts due to varying thicknesses being misinterpreted as dislocations more than 50 years ago!

Furthermore the paragraph starting "Notably, not all..." that they added in response to referee #2 is horribly wrong. The authors seem to believe that diffraction spots change position with the excitation error. This is completely wrong, it depends upon the combination of the excitation error and the surface normals. No paper should be published with this text, it will move science backwards. While the book by Eddington is useful, it is not a high level text. The authors need to read and understand topics such as diffractive multiplets as discussed in Cowley's book; they should also read and understand the book by Peng, Dudarev and Whelan is a more modern text, also probably harder to understand as it goes deeper.

My opinion is now a definitive reject, and send to Acta Cryst.

Answer: Thank you for your comments. The last revision was indeed not straightforward and clear enough in the introductory part. In the current revision, we made further modifications to improve the manuscript.

The reviewer #4 may not follow our previous discussions with reviewer #2. We were interpreting

the diffraction phenomenon under SAED mode using parallel beam condition. It is certain that the dislocation affects the position of the diffraction spots as described in the book “Diffraction Physics” by Cowley and in “Practical Electron Microscopy in Materials Science” written by Edington. For columns passing through the dislocation core, the planes of atoms are displaced as at a stacking fault and the main effect of the structure away from the dislocation is to tilt the lattice planes towards or away from the Bragg angle, resulting the position change of the diffraction spots. This phenomenon can be observed using the nanobeam mode or two-beam conditions, however, the SAED method is not sensitive to the detailed configuration of atoms around the dislocation core. Therefore, our statement “the dislocation should only influence the shape of diffraction spots without causing the deviation in the Bragg reflection position” discussed the 3D ED data collection under the SAED mode.

Regarding the excitation error, the reciprocal lattice points are elongated perpendicular to the crystal surface and become reciprocal-lattice rods through the effect of the shape function. The Ewald sphere is tangent to different positions of the elongated diffraction spots, which leads to excitation error. In the data acquisition process using 3D EDT, all the existing diffraction intensity in the 3D reciprocal space within the corresponding goniometer rotation range can be covered and reconstructed. Therefore, the excitation error due to the curvature of the Ewald sphere may not be the key to the deviation from the exact Bragg position in our experiment.

After carefully considering the experimental data, we found that the deviation in the position of the diffraction spots is corresponding to the crystal lattice torsion in the chiral sample. The continuous rotation of the reflecting planes in the chiral sample changes the value of deviation from the Bragg reflection position, which varies from point to point in the specimen. This twisting geometry leads to the continuous shifting of the diffraction spots, particularly for the high-order diffraction spots. We have added a new Supplementary Figure 27 to illustrate this phenomenon, which is highly consistent with the observed position changes in the experiment (the yellow arrow indicates the deviation direction of diffraction). The simulation results also verified this geometric relationship. We also referred the book of “High Energy Electron Diffraction and Microscopy” by Peng, Dudarev, and Whelan. This book states that effects of multiple scattering by fluctuations often make a very significant contribution to the distribution of intensity in electron diffraction patterns. Close to the surface of the crystal, the potential periodicity may be influenced by the multiple diffraction scattering due to surface roughness or the presence of a disordered layer of nanosheets segregated at the surface, resulting in the deviations of the potential from the dynamic (phonons or electronic excitations).

According to these discussions, we have revised our text into “Interestingly, the high-order diffraction spots are shifted from the original reciprocal lattice sites in 3D ED data (Supplementary Fig. 26), which can be interpreted by the deviation from the Bragg reflection position due to the twisting of chiral crystal with continuous distortion of lattice planes (Supplementary Fig. 27). This phenomenon may also relate to the change of diffraction intensity distribution induced by local thickness variations and the effect of multiple scattering.”

Supplementary Figure S27. Schematic drawing and simulation of the deviation of the diffraction spots from the reciprocal lattice points induced by the crystal lattice torsion. **a**, The modulated 3D ED model of the twisted crystal. The models were built by rotating the initial lattice of a single crystal by two perpendicular axes corresponding to lattice torsion with small angular steps. The yellow and red dots represent the centers of the first and last arched-shape diffraction in the series of modulated diffraction spots. **b**, The enlargement of a column of high-order arch-shaped diffraction spots (marked by the grey box in **a**). The red dots represent the primary reciprocal lattice sites on the red perpendicular reference lines and the blue dots represent the center of the modulated arch-shaped diffraction. The yellow arrow indicates the direction of the diffraction spots shifting caused by the torsional crystal lattice. **c**, The simulation of 3D ED patterns of the twisted CTD crystal. The yellow and red dots represent the center of the first and last diffraction on a series of arched-shaped reciprocal-lattice rods. The positional deviations between yellow and red dots can be observed due to the twisting of chiral crystal. In this simulation, eighteen tin dioxide nanocrystals are twisted by 0.6° for each crystal along the direction perpendicular to the $(1\bar{1}0)$ plane.